# Recent Development in Vanadium Pentoxide and Carbon Hybrid Active Materials for Energy Storage Devices

**DOI:** 10.3390/nano11123213

**Published:** 2021-11-26

**Authors:** Andrew Kim, Golap Kalita, Jong Hak Kim, Rajkumar Patel

**Affiliations:** 1Department of Chemical Engineering, The Cooper Union for the Advancement of Science and Art, New York, NY 10003, USA; kim70@cooper.edu; 2Department of Physical Science and Engineering, Nagoya Institute of Technology, Gokiso-Cho, Showa-ku, Nagoya 466-8555, Japan; golapkalita@gmail.com; 3Department of Chemical and Biomolecular Engineering, Yonsei University, 50, Yonsei-ro, Seodaemun-gu, Seoul 03722, Korea; 4Energy & Environmental Science and Engineering (EESE), Integrated Science and Engineering Division (ISED), Underwood International College, Yonsei University, 85 Songdogwahak-ro, Yeonsugu, Incheon 21983, Korea

**Keywords:** electrochemical energy storage, supercapacitor, vanadium pentoxide, carbon nanocomposite

## Abstract

With the increasing energy demand for portable electronics, electric vehicles, and green energy storage solutions, the development of high-performance supercapacitors has been at the forefront of energy storage and conversion research. In the past decade, many scientific publications have been dedicated to designing hybrid electrode materials composed of vanadium pentoxide (V_2_O_5_) and carbon nanomaterials to bridge the gap in energy and power of traditional batteries and capacitors. V_2_O_5_ is a promising electrode material owing to its natural abundance, nontoxicity, and high capacitive potential. However, bulk V_2_O_5_ is limited by poor conductivity, low porosity, and dissolution during charge/discharge cycles. To overcome the limitations of V_2_O_5_, many researchers have incorporated common carbon nanostructures such as reduced graphene oxides, carbon nanotubes, carbon nanofibers, and other carbon moieties into V_2_O_5_. The carbon components facilitate electron mobility and act as porous templates for V_2_O_5_ nucleation with an enhanced surface area as well as interconnected surface morphology and structural stability. This review discusses the development of various V_2_O_5_/carbon hybrid materials, focusing on the effects of different synthesis methods, V_2_O_5_/carbon compositions, and physical treatment strategies on the structure and electrochemical performance of the composite material as promising supercapacitor electrodes.

## 1. Introduction

The demand for improved energy storage devices has increased due to the rapid development of portable electronics, electric vehicles, and green energy storage devices [1]. Supercapacitors are promising replacements for traditional energy storage devices such as batteries and capacitors with high energy and power densities, respectively, because supercapacitors can be fabricated using readily accessible materials with outstanding cyclability and can provide a balance of both high power and energy densities (Figure 1) [2,3,4,5]. The two main classifications of supercapacitors are electric double-layer capacitors (EDLCs) and pseudocapacitors (faradic supercapacitors) [6,7]. EDLCs produce a charge separation at the boundary between the electrode and electrolyte to store energy [8]. In contrast, pseudocapacitors rely on fast faradic reactions at the electrode surface to store energy [9]. Depending on the electrode material, supercapacitors primarily exhibit electric double-layer (EDL) or pseudocapacitive characteristics or a combination of both.

Transition metal oxide/carbon hybrid materials have recently attracted considerable attention as composites with easily controllable pseudocapacitive and EDL characteristics. Transition metal oxides such as vanadium pentoxide (V_2_O_5_) are pseudocapacitive, yielding supercapacitors with high specific capacitances (C_sp_) and energy densities [10]. However, V_2_O_5_ is limited by poor conductivity, low power density, and minimal cyclic stability [11]. In contrast, carbon nanostructures, such as reduced graphene oxide (rGO), carbon nanotubes (CNTs), and carbon nanofibers (CNFs), show dominant EDL characteristics, resulting in highly stable and power-dense supercapacitors. However, carbon-based materials suffer from low specific capacitances and energy densities [12]. Transition metal oxide/carbon composite materials supplement the high energy potential of transition metal oxides with the high power potential and stability of carbon nanostructures [13,14,15]. These composite materials will be essential to meet the demand for fast-charging portable electronics, long-lasting electric vehicles, and environmentally friendly energy storage devices.

V_2_O_5_ has attracted significant attention as a transition metal oxide with multiple oxidation states (II–V), enabling a high maximum theoretical capacitance of 2120 F g^−1^ [16]. With a wide effective potential window, V_2_O_5_ can provide a high theoretical energy density [17]. Additionally, owing to its natural abundance and low toxicity, V_2_O_5_ is a low-cost material that is ideal for mass production [18,19]. The crystal structure of V_2_O_5_ allows electrolyte ions such as Li^+^ to reversibly intercalate/de-intercalate, thereby improving the faradic reactivity with the electrolyte [20,21]. However, bulk V_2_O_5_ is limited by poor electrical conductivity, slow reaction kinetics, and vanadium dissolution [22,23]. To improve the properties of bulk V_2_O_5_, the bulk crystal nanostructure has been converted into nanorods [24], nanotubes [25], nanosheets [26], nanobelts [27], and other porous nanostructures [28]. These nanostructures allow better reaction kinetics with shorter diffusion pathways than those observed in the bulk crystal structure, and they improve cyclability with less strain on the crystal structure during ion intercalation/de-intercalation. However, these V_2_O_5_ materials still have drawbacks, such as low electrical conductivity [29,30].

Recent developments in improving V_2_O_5_ supercapacitor materials involve the addition of carbon nanomaterials, such as graphene [31], rGO [32], CNTs [33], and activated carbon (AC) [34], to improve the conductivity and structural stability of V_2_O_5_. Carbon materials are ideal sources of EDL capacitance for efficient supercapacitors, owing to high porosities, conductivities, and natural abundances [35,36]. Highly porous carbon materials have large surface areas, resulting in more active sites on the electrode material [37]. Increased porosity allows fast ion intercalation/de-intercalation with short ion diffusion pathways [38]. The large surface area improves the interfacial contact between the electroactive material and the current collector, resulting in more efficient electron transfer. Moreover, the carbon content improves the overall conductivity of the electrode, increasing the specific capacitance and decreasing the energy losses during charge/discharge [39].

Despite the recent developments in V_2_O_5_/carbon composites, many review articles on supercapacitor electrode materials only briefly discuss V_2_O_5_ materials as part of a broader review of transition metal oxide [40,41,42,43,44,45] or carbon-based supercapacitors [46,47,48]. V_2_O_5_/carbon hybrid materials are promising materials that have been the focus of recent research, so it is essential to organize the most up-to-date information on factors affecting the performances of V_2_O_5_/carbon composite electrodes. This review describes the physical and electrochemical characteristics of different V_2_O_5_/carbon nanostructures, including V_2_O_5_/rGO, V_2_O_5_/CNTs, V_2_O_5_/CNFs, and other V_2_O_5_/carbon hybrid materials. This review focuses on the effects of different synthesis methods, carbon to V_2_O_5_ ratios, and physical treatment procedures on the structures and performances of V_2_O and carbon hybrid nanomaterials.

## 2. V_2_O_5_/rGO

rGOs have been extensively investigated as additive materials for V_2_O_5_ composites because of their large surface areas, high conductivities, and good stabilities [49]. Similar to pristine graphene (sometimes reported in the literature as graphene), rGO is a monolayer of sp^2^ hybridized carbon atoms arranged in a hexagonal lattice [50,51]. However, unlike pure graphene, rGO contains varying degrees of defects caused by functional groups such as hydroxyl and carbonyl groups remaining after the reduction of highly functionalized graphene oxide (GO) [52]. Various factors involved in the fabrication of a V_2_O_5_/rGO (VrG) electrode affect its electrochemical performance by altering the morphology and crystalline structure of the hybrid material. VrG composites are versatile materials that often possess a lamellar structure with high porosity and surface area. The effects of different synthesis pathways, V_2_O_5_/carbon compositions, and physical treatment conditions were closely examined for their impact on the nanostructures and the resulting capacitive performances of the VrG electrodes. The morphology and electrochemical performances of V_2_O_5_/rGO electrodes for supercapacitor applications reported in the literature are summarized in Table 1.

### 2.1. Effects of Synthesis Method

Many synthesis strategies for the fabrication of VrG composites include a hydro/solvothermal method. Typically, a mixture of V_2_O_5_ precursors such as vanadium oxytriisopropoxide (VTIP), rGO precursors such as GO, and water or other solvents are heated at high temperatures and pressures for extended periods in a Teflon-lined stainless steel autoclave [53]. The hydro/solvothermal procedure is frequently used because it is facile and allows the formation of diverse V_2_O_5_ morphologies on the 2D rGO substrate. Pandey et al. synthesized V_2_O_5_ nanospheres anchored to thin rGO sheets via a hydrothermal synthesis route. A uniform dispersion of VTIP, GO, isopropyl alcohol, and DI water was heated at 180 °C in an autoclave for 18 h, yielding a mesoporous VrG composite material [54]. The resulting material had a layered structure with large V_2_O_5_ nanospheres intercalated into the rGO layers. The vanadium crystals had an organized orthorhombic crystal structure that promoted deep ion adsorption. The lamellar structure of rGO increased the surface area for additional surface redox reactions and porosity for abundant electrolyte ion intercalation/de-intercalation. In a symmetric, two-electrode configuration with the VrG working electrodes, the composite exhibited a large maximum C_sp_ of 448 F g^−1^ at a current density of 0.75 A g^−1^ that decreased slightly to 296 F g^−1^ at a significantly higher current density of 15.5 A g^−1^. The excellent rate capability was due to the low charge transfer resistance (0.6 Ω), which was enabled by the strong bonds formed between the intercalated V_2_O_5_ nanospheres and conductive rGO sheets.

The hydro/solvothermal synthesis route can also yield V_2_O_5_ nanowires. Ahirrao et al. followed a similar hydrothermal process with ammonium metavanadate (NH_4_VO_3_) to yield a similar lamellar rGO structure with intercalated V_2_O_5_ nanowires [55]. First, NH_4_VO_3_ was calcinated into V_2_O_5_, and the V_2_O_5_ powder was then sonicated with GO in DI water and heated at 180 °C for two days in an autoclave. The hydrothermal processes yielded thin nanowires with lengths ranging from 100 nm to several micrometers anchored to the rGO layers. The nanowire morphology increased the surface area of V_2_O_5_ for more active sites and decreased the ion diffusion pathways. The VrG composite was drop-cast onto carbon paper to yield a supercapacitor electrode. The VrG electrode exhibited a maximum C_sp_ of 1002 F g^−1^ at a current density of 1 A g^−1^. Because of its low charge transfer resistance (0.53 Ω), the composite exhibited good rate capability, as indicated by the high C_sp_ of 828 F g^−1^ at a high current density of 3 A g^−1^. Similarly, Geng et al. synthesized thin V_2_O_5_ nanowires anchored to curly rGO sheets using a hydrothermal process [56]. As the hydro/solvothermal process often yields a VrG composite powder, it is combined with a conducting filler and polymer binder and coated onto a current collector for use as an electrode. The VrG material was combined with Super P carbon (conducting filler), polyvinylidene fluoride (PVDF) (polymer binder), and N-methyl-2-pyrrolidone (NMP) (solvent) in an 8:1:1 ratio. The resulting slurry was coated onto Ni foam, vacuum dried, and compressed into a thin sheet. Unlike a 2D metal foil current collector, Ni foam provided a 3D macroporous network for large volumes of electrolyte diffusion. Anchoring the VrG material to the porous foam increased the surface area for more active redox sites. The electrochemical performance of the VrG hybrid material was tested in a three-electrode configuration with a VrG working electrode, Pt counter electrode, and Ag/AgCl reference electrode in 1 M KCl electrolyte. The Ni foam-based VrG electrode exhibited a high C_sp_ of 579 F g^−1^ at a current density of 1 A g^−1^. Despite the lower conductivity than that of Cu, the relatively high conductivity and larger pore size of Ni foam enabled good rate capability, as indicated by a large C_sp_ of 534 F g^−1^ at a high current density of 4 A g^−1^. Because of its rigid and porous structure, the VrG material was less susceptible to mechanical strain arising from electrolyte intercalation/de-intercalation, resulting in a high C_sp_ retention of 79% after 5000 cycles at a current density of 4 A g^−1^. Sun et al. found that the duration of the hydro/solvothermal reaction affected the growth of V_2_O_5_ nanowires on rGO sheets [57]. V_2_O_5_ was modified with the N-doped rGO (N-rGO) aerogel via solvothermal synthesis in an autoclave at 160 °C for different durations. The composite was then freeze-dried to yield a marshmallow-shaped, free-standing VrG electrode. Tiny hair-like V_2_O_5_ nanowires were vertically anchored on the surfaces of the amorphous N-rGO sheets (Figure 2).

A short reaction time of 30 min between V_2_O_5_ and rGO resulted in a smooth rGO surface without many V_2_O_5_ nanowire growths (Figure 2a). After 45 to 60 min, larger V_2_O_5_ nanowires were obtained (Figure 2b,c), suggesting additional V_2_O_5_ nucleation and growth. Prolonged reactions for 90 to 120 min resulted in long, vertical V_2_O_5_ crystal growths uniformly spread on the rGO surface (Figure 2d,e). A magnified view of the V_2_O_5_ coated surface (Figure 2f) showed the porosity of vertically grown V_2_O_5_ nanowires. The mesoporous network of N-rGO sheets combined with the nanowire extrusions resulted in a large BET surface area of 416 m^2^ g^−1^, which allowed fast ion diffusion and increased the number of active sites for surface reactions. The porous VrG electrode exhibited an excellent C_sp_ of 710 F g^−1^ at a current density of 0.5 A g^−1^. The nitrogen defects in the rGO frame lowered the charge transfer resistance of the electrode from 1.54 to 1.11 Ω, resulting in improved rate capability, indicated by a high C_sp_ of 360 F g^−1^ at a large current density of 10 A g^−1^. As the nitrogen groups in rGO were more reactive than carbon, N-rGO contained more V_2_O_5_ nucleation sites than bare rGO, resulting in denser distributions of V_2_O_5_ nanowires throughout the N-rGO-based composite. The V_2_O_5_/N-rGO material exhibited excellent cyclic stability with 95% C_sp_ retention after 20,000 cycles at a high current density of 10 A g^−1^.

In addition to nanospheres and nanowires, the hydrothermal process can yield more sheet-like V_2_O_5_ morphologies anchored to rGO. Nagaraju et al. synthesized a VrG composite with V_2_O_5_ and rGO nanosheets via a hydrothermal method to obtain a lamellar composite structure [58]. V_2_O_5_ had a pure orthorhombic crystal structure that allowed good ion diffusion. The layered structure of the V_2_O_5_ and rGO nanosheets resulted in a large surface area of 36.2 m^2^ g^−1^, which was four times larger than that of bulk V_2_O_5_. The VrG material exhibited both EDL and pseudocapacitive characteristics because of the even composition of V_2_O_5_ and rGO. The porosity and crystallinity enhanced the pseudocapacitive capability to a maximum C_sp_ of 635 F g^−1^ at a current density of 1 A g^−1^. Even at a high current density of 30 A g^−1^, the composite exhibited a large 240 F g^−1^ C_sp_, highlighting the layered composite’s potential for high-energy and high-power applications. An asymmetric supercapacitor with the VrG electrode exhibited an energy density of 75.9 Wh kg^−1^ with a power density of 900 W kg^−1^. Sahu et al. used a hydrothermal method to synthesize V_2_O_5_ nanostrips anchored to rGO nanoribbon [59]. The V_2_O_5_ nanostrips shrank during the hydrothermal reaction with rGO as the rGO template inhibited large V_2_O_5_ growth. The retention of small V_2_O_5_ crystals doubled the surface area to 15.6 m^2^ g^−1^, thereby increasing the efficiencies of surface redox reactions. The composite material had high mesoporosity, facilitating the fast diffusion of electrolytes through the ion channels. Thus, the C_sp_ of the composite (309 F g^−1^) was approximately five times greater than that of bulk V_2_O_5_ nanostrips. The VrG material had a low equivalent series resistance and charge transfer resistance of 4.6 and 1.2 Ω, respectively, which provided good rate capability, indicated by the relatively high 114 F g^−1^ C_sp_ at a high scan rate of 100 mV s^−1^. A solid-state electrode comprising the VrG composite exhibited a high conductivity of 1.4 × 10^−2^ S m^−1^, which minimized the energy loss as heat. A symmetric VrG electrode supercapacitor had an energy density of 42.09 Wh kg^−1^ at a power density of 475 W kg^−1^, which decreased to 13.44 Wh kg^−1^ at a power density of 8400 W kg^−1^.

The facile hydrothermal method can also alter the large sheet-like structure of rGO. Zhang et al. employed a hydrothermal and freeze-drying procedure to fabricate a VrG hydrogel with thin rGO strips instead of large sheets [60]. The outer appearance of the hydrogel constituted a marshmallow-like macrostructure. The rGO strips were laced together to form a macroporous 3D structure. The V_2_O_5_ nanobelts were uniformly intercalated in the web-like rGO architecture. The VrG electrode with 64 wt% V_2_O_5_ exhibited a large C_sp_ of 320 F g^−1^ at a current density of 1.0 A g^−1^. In a symmetric supercapacitor setup, VrG retained 70% of its initial C_sp_ after 1000 cycles at a current density of 1 A g^−1^.

An alternative to the versatile hydrothermal process is the sol-gel method, which is a low-cost synthesis strategy that involves the formation of a V_2_O_5_ sol that is converted into a gel via hydrolysis and condensation reactions. The resulting porous V_2_O_5_ xerogel can be used as a free-standing, binderless material used directly as a supercapacitor electrode [61]. Yilmaz et al. utilized the sol-gel synthesis route to fabricate free-standing VrG electrodes [62]. The V_2_O_5_ gel was first synthesized via a hydrothermal process involving V_2_O_5_ powder and H_2_O_2_. The V_2_O_5_ gel, GO, and thiourea (cross-linking agent) were reacted for two weeks in a cylindrical glass vial. The resulting VrG aerogel was washed with ethanol, freeze-dried, and annealed at 300 °C in air. The highly porous composite had a large BET surface area of 83.4 g m^2^ because of the lamellar rGO sheet structure with V_2_O_5_ nanoribbons anchored to the rGO surface. The addition of thiourea increased the chemical grafting between rGO and V_2_O_5_ by functioning as a redox couple, temporarily reducing V^5+^ to V^4+^ to initiate the polymerization between rGO and V_2_O_5_. The reduced V^4+^ was subsequently oxidized to V^5+^ during the final annealing step, yielding a hydrated orthorhombic V_2_O_5_ crystal structure. The symmetric supercapacitor with the thiourea-doped VrG electrodes exhibited a C_sp_ of 484.9 F g^−1^ at a current density of 0.6 A g^−1^, which was twice that of a thiourea-less VrG. Using thiourea during the synthesis also resulted in sulfur and nitrogen functionalization of rGO. The functional groups decreased the equivalent series resistance to 1.6 Ω, resulting in good rate capability with a high C_sp_ of ~300 F g^−1^ at a high current density of 10 A g^−1^. The mesoporous structure allowed high ion adsorption and decreased the mechanical strain during rapid ion intercalation/de-intercalation, resulting in a high C_sp_ retention of 80% after 10,000 cycles at a high current density of 5 A g^−1^. A symmetric capacitor with the VrG xerogel electrodes possessed an energy density of 43.0 Wh kg^−1^ at a power density of 480 W kg^−1^, which decreased to 24.2 Wh kg^−1^ at a power density of 9300 W kg^−1^. Kiruthiga et al. also used the sol-gel method to synthesize V_2_O_5_ nanorods anchored to rGO sheets [63]. A V_2_O_5_ sol was prepared by reacting ammonium metavanadate with citric acid, and the sol was heated to form a gel that was subsequently heated at 450 °C in air to produce V_2_O_5_ powder. V_2_O_5_ was subsequently intercalated into the rGO sheets via sonication in DI water. The VrG composite exhibited a C_sp_ of 224 F g^−1^ at a current density of 0.01 A g^−1^ with a high C_sp_ retention of 85% after 1000 cycles at a current density of 0.06 A g^−1^. An asymmetric Na-ion supercapacitor with a VrG anode and an AC cathode exhibited a maximum C_sp_ of 62 F g^−1^ at a current density of 0.01 A g^−1^ with 74% C_sp_ retention after 1000 cycles at a current density of 0.06 A g^−1^. The supercapacitor produced a maximum energy density of 65 Wh kg^−1^ at a power density of 72 W kg^−1^ and a current density of 0.03 A g^−1^.

Another method of synthesizing a VrG composite involves the filtration of an rGO and V_2_O_5_ suspension through a membrane. Unlike the hydrothermal method, the filtration process produces a stable and binderless thin-film electrode material. Wang et al. synthesized a VrG composite via the filtration of a solution of poly(3,4-ethylenedioxythiophene) (PEDOT), a conducting polymer, V_2_O_5_, and rGO through a cellulose acetate membrane [64]. The resulting binderless VrG thin film was then roll pressed onto various substrates such as ITO glass and Al to yield a substrate/PEDOT/VrG electrode. During synthesis, the V_2_O_5_ crystals were recrystallized into thin nanobelts coated evenly with PEDOT without the intercalation of the 3,4-ethylenedioxythiophene (EDOT) monomer into the V_2_O_5_ lattice. PEDOT acted as a bridge between V_2_O_5_ and rGO through π–π conjugation, such that V_2_O_5_ did not interact directly with rGO. These strong bonds allowed the composite material to form a stable film structure that was easily transferred to and compressed onto different substrates without a binder between the substrate and composite material. Figure 3a shows the procedure for transferring the thin-film hybrid material. A facile electrode fabrication procedure was applied to various substrates such as ITO, plastic, and glass. The VrG material exhibited smooth surface adhesion and good optical transparency for all tested substrates (Figure 3b,c). A large thin-film electrode with a diameter of 170 mm was easily fabricated (Figure 3d) and placed in an array (Figure 3e), demonstrating the viability of this fabrication strategy for scale-up. When placed on an Ag/PET current collector, the PEDOT/VrG electrode yielded an areal capacitance of 22.4 mF cm^−2^ at a current density of 0.7 A m^−2^. A lack of binder between the substrate and PEDOT/VrG enabled direct contact between the conductive Ag substrate and composite material, increasing the overall electrochemical performance. The PEDOT coating on V_2_O_5_ reduced the dissolution of V_2_O_5_, resulting in a high capacitance retention of 98% after 150,000 cycles. Moreover, the outer rGO layer trapped the vanadium ions within the composite structure for maximum vanadium retention. A symmetric supercapacitor yielded a high C_sp_ retention of 92.4% after 50,000 cycles with a maximum energy density of 1800 Wh m^−2^ at a power density of 110,000 W m^−2^. Similarly, Liu et al. synthesized a stable V_2_O_5_/rGO composite electrode via vacuum filtration through a cellulose acetate membrane [65]. A thin VrG film was fabricated by first preparing a lytropic liquid crystal suspension of rGO and V_2_O_5_ in DI water. The lamellar phase suspension was filtered through the cellulose membrane, yielding a VrG thin film that could withstand a maximum pressure of 120 MPa. With a 67 wt% V_2_O_5_, the thin film possessed a maximum C_sp_ of 205 F g^−1^ at a current density of 1 A g^−1^. The electrode exhibited excellent rate capability by retaining more than 50% of its initial C_sp_ at a high current density of 50 A g^−1^. The electrode retained 94% of its C_sp_ after 3000 cycles at a current density of 10 A g^−1^. An added advantage of thin-film VrG materials is their high flexibility. Foo et al. utilized vacuum filtration through a nitrocellulose filter to synthesize a flexible VrG composite [66]. The VrG was peeled from the filter, heated in an autoclave with hydrazine monohydrate, dried, and acid-treated to yield a flexible, binderless VrG electrode. The hydrazine exfoliated the VrG into layers with an average spacing of 30 μm. Small amorphous V_2_O_5_ crystals were embedded on the surface of planar rGO layers. The thin-film material possessed a large Young’s modulus of 1.7 GPa and tensile strength of 6.1 MPa and could be repeatedly bent and unbent around a test tube with no sign of permanent deformation. The flexible electrode material exhibited a C_sp_ of 178.5 F g^−1^ at a current density of 0.05 A g^−1^. The hybrid thin film had a moderate rate capability due to a low equivalent series resistance of 3.36 Ω, resulting in a C_sp_ of 129.7 F g^−1^ at twice the current density. An asymmetric supercapacitor with a flexible VrG anode had maximum energy densities of 13.3 Wh kg^−1^ (unbent) and 13.6 Wh kg^−1^ (bent) at a power density of 12.5 W kg^−1^.

Chemical deposition is a facile strategy for synthesizing a VrG composite directly on a conductive substrate. Different deposition techniques allow the composite material to bind to a well-structured, porous 3D network for structural stability and maximum porosity without the need for insulative additives or binders. Van Hoa et al. utilized chemical vapor deposition (CVD) to fabricate a free-standing graphene/V_2_O_5_ composite with a Ni foam template [67]. First, graphene was deposited on a Ni foam template using CVD, yielding a porous macrostructure with smooth graphene plates. Subsequently, V_2_O_5_ was deposited on the graphene/Ni foam substrate via a solvothermal method in an oxalic acid solution. The Ni foam was covered with uneven graphene plates after the initial CVD, increasing the surface area of the template. Small V_2_O_5_ nanoflowers were uniformly packed on the graphene surface. The flower-like nanostructures were intricately connected ultrathin V_2_O_5_ nanosheets with a highly orthorhombic crystal phase. The nanoflower extrusions on the porous graphene/Ni foam template further increased the BET surface area to 49.4 m^2^ g^−1^ for increased redox reactions. As V_2_O_5_ was synthesized directly on the graphene/Ni foam template, there was no binder material to reduce the number of active sites. The electrode exhibited one of the highest reported C_sp_ values of 1235 F g^−1^ at a current density of 2 A g^−1^. The direct contact between graphene and Ni foam resulted in high conductivity throughout the electrode material, resulting in a slight decrease in C_sp_ to 800 F g^−1^, even at a large current density of 20 A g^−1^. The large capacitance was due to the high specific surface area provided by the porous template and gaps between the petal-like V_2_O_5_. The composite electrode retained 92% of its C_sp_ after 5000 cycles at a high current density of 4 A g^−1^, indicating good reversibility because of stable chemical bonding and enhanced conductivity. The energy density was 116 Wh kg^−1^ at a power density of 440 W kg^−1^, which decreased to ~330 Wh kg^−1^ at a power density of ~3500 W kg^−1^. Wang et al. employed a similar synthesis strategy to fabricate a Ni foam-based V_2_O_5_/rGO electrode [68]. HCl and FeCl_3_ were used to etch the Ni foam before graphene CVD to ensure strong, more homogeneously spread bonding. V_2_O_5_ was then synthesized directly on the rGO/Ni foam substrate via a solvothermal method in an ethanol solution, resulting in long V_2_O_5_ nanoribbons that were several micrometers in length. V_2_O_5_ was not completely oxidized and possessed a monoclinic crystal phase, while rGO showed a dominant graphitic crystal phase. The areal capacitance of the free-standing electrode was 822 mF cm^−2^ at a current density of 1 mA cm^−2^. An asymmetric supercapacitor composed of a VG cathode had an energy density of 16 Wh kg^−1^ at a power density of 200 W kg^−1^. Using chemical bath deposition (CBD) instead of CVD, Korkmaz et al. synthesized a binder-free, thin-film VrG [69]. Glass (G), poly(methyl methacrylate) (PMMA) (P), fluorine-doped tin oxide (FTO) glass (F), and indium tin oxide (ITO) glass (I) substrates were added to a solution of GO, NaVO_3_, galactic acid, and methanol, and reacted for 24 h at ambient temperature. The coated substrate materials were removed via chemical blanket removal. The texture of the film surface differed for each substrate, showing spherical structures of different sizes for VrG-G, a smooth and homogeneous structure for VrG-P, a dense and grainy structure for VrG-F, and dispersed agglomerations for VrG-I. The crystallinity also changed depending on the substrate with amorphous, orthorhombic crystal growth observed in VrG-G and VrG-P, but well-oriented orthorhombic crystal growth in VrG-F and VrG-I. The thickness of the VrG film was also different, with values of 1025, 988, 689, and 393 nm for VrG-G, VrG-P, VrG-F, and VrG-I, respectively. Despite containing a thin V_2_O_5_ layer for limited pseudocapacitance potential, VrG-F exhibited the largest C_sp_ of 949.6 F g^−1^ owing to its organized crystalline structure.

Different synthesis strategies such as hydro/solvothermal, sol-gel, filtration, and chemical deposition methods are available for the preparation of VrG composites with varying V_2_O_5_ and rGO morphologies. The hydrothermal method is frequently used as a facile and versatile means of embedding V_2_O_5_ nanospheres, nanowires, nanorods, and nanosheets onto lamellar rGO sheets. The product is typically a VrG powder that must be combined with a conductive filler and binder for use as an electrode. The sol-gel method is an alternate process that yields a porous gel instead of a powder that can be directly applied as a supercapacitor electrode. The filtration method yields a thin-film, binderless electrode that can be physically transferred to different substrates such as ITO and plastic. Chemical deposition techniques such as CVD and CBD also allow the direct growth of V_2_O_5_ on porous and conductive substrates such as Ni foam and ITO for improved conductivity and stability.

### 2.2. Effects of Composition

Altering the ratio of rGO and V_2_O_5_ can improve the overall electrochemical performance by balancing the conductivity of rGO and the pseudocapacitance of V_2_O_5_. Li et al. emphasized the importance of carefully tuning the carbon content for improved electrochemical performance [70]. Different amounts of rGO were added during hydrothermal synthesis to yield VrG electrodes with 15, 22, and 26 wt% rGO. Even a relatively small difference of 4 wt% in the carbon content between VrG-22 and VrG-26 resulted in a significant increase in conductivity, with VrG-26 retaining 46% of its charge capacity (compared to 26% capacity retention of VrG-22) when the scan rate was increased from 1 to 20 mV s^−1^. As the orthorhombic V_2_O_5_ nanorods were wrapped by and intercalated between the crumpled rGO sheets, an increase in the rGO content resulted in few nanorod agglomerations and high conductivity, resulting in improved rate capability. An asymmetric capacitor with a VrG-26 anode and AC cathode exhibited a C_sp_ of 37.2 F g^−1^ at a current density of 0.5 A g^−1^. The integrated, layered structure of rGO reduced the permanent dissolution of V_2_O_5_, leading to 90% C_sp_ retention after 1000 cycles at a current density of 2 A g^−1^. The maximum energy density exhibited was 54.3 Wh kg^−1^ at a power density of 136.4 W kg^−1^. Saravanakumar et al. physically mixed different ratios of already prepared rGO and V_2_O_5_ in disodium citrate and aged the solution for three days to yield a VrG composite [71]. All VrG compositions had an orthorhombic V_2_O_5_ crystal structure and produced a complex V_2_O_5_ network on the 2D rGO surface. However, among these, VrG-5 (5 wt% rGO) and VrG-15 (15 wt% rGO) resulted in more agglomeration of V_2_O_5_, whereas VrG-10 (10 wt% rGO) had a more homogeneous spread of the V_2_O_5_ crystals. V_2_O_5_ agglomeration limited surface ion adsorption as the electrolyte could not intercalate deep into the bulk V_2_O_5_ crystal structure. Thus, well-spread VrG-10 exhibited the highest C_sp_ of 519 F g^−1^ at a scan rate of 2 mV s^−1^, which was significantly greater than the 326 F g^−1^ C_sp_ for pristine V_2_O_5_. An increase in rGO content improved the electron mobility to balance the high pseudocapacitive charge capacity of V_2_O_5_. VrG-15 performed worse than bare V_2_O_5_ with a C_sp_ of 210 F g^−1^ because of the lower V_2_O_5_ content and V_2_O_5_ agglomeration on rGO, which limited electron mobility and ion intercalation/de-intercalation. An increase in the rGO content enhanced the rate capability of the VrG electrodes, with VrG-5 showing only 47% C_sp_ retention compared to 68% retained by VrG-15 when the current density was increased from 0.5 to 10 A g^−1^. Increasing the rGO content decreased the internal resistance, as indicated by 0.4 Ω resistance in VrG-10 compared to 1.73 Ω for pristine V_2_O_5_. The physical support provided by the rGO sheets further reduced the damage caused by repeated intercalation/de-intercalation, resulting in an 83% C_sp_ retention after 1000 cycles for VrG-10. An appropriate rGO composition is required to decrease V_2_O_5_ agglomeration for maximum pseudocapacitive efficiency and improve the conductive pathways for higher rate capability.

Although a high carbon content is desirable, a high V_2_O_5_ composition is required for energy-dense supercapacitor electrodes. Ramadoss et al. examined the effect of varying rGO to V_2_O_5_ ratio by controlling the initial ratio of GO and V_2_O_5_ during microwave synthesis [72]. All three VrG variations, V_1_rG_2_ (1:2 V_2_O_5_ to GO), V_1_rG_1_ (1:1 V_2_O_5_ to GO), and V_2_rG_1_ (2:1 V_2_O_5_ to GO), possessed a pure orthorhombic crystal phase with uniform nanorods of 150–200 nm lengths. The VrG electrodes exhibited mixed pseudocapacitive and EDL contributions to the overall capacitance. An increase in the V_2_O_5_ content increased the C_sp_ of the VrG electrode, as indicated by the higher C_sp_ of 250 F g^−1^ for V_2_rG_1_ than 103 F g^−1^ for V_1_rG_2_ at a scan rate of 5 mV s^−1^. As V_2_O_5_ contributed toward most of the capacitance via faradic reactions, increasing the proportion of V_2_O_5_ increased the total capacitance. A symmetric supercapacitor with V_2_rG_1_ electrodes had an energy density of 12.5 Wh kg^−1^ at a power density of 79,900 W kg^−1^, which decreased to 8.4 Wh kg^−1^ at 10 times the power density. Lee et al. also concluded that high V_2_O_5_ content increased the total capacitance [73]. VrG material was synthesized via a low-temperature hydrothermal process with different ratios of V_2_O_5_ powder and GO. V_3_rG_1_ (3:1 V_2_O_5_ to GO), V_1_rG_1_ (1:1 V_2_O_5_ to GO), and V_1_rG_3_ (1:3 V_2_O_5_ to GO) all yielded orthorhombic crystalline V_2_O_5_ nanobelts with uniform size. V_2_O_5_-rich V_3_rG_1_ outperformed the other two variations with a C_sp_ of 288 F g^−1^ at a scan rate of 10 mV s^−1^, doubling the C_sp_ of V_1_rG_3_. Although an increase in the rGO content theoretically increased the conductivity, the low-temperature hydrothermal reduction led to only a partial reduction of GO to rGO, which limited the conversion to highly conductive rGO [73]. A large capacitance requires significant pseudocapacitive contributions, supporting the need for a V_2_O_5_-rich composite material. Many active sites were available for surface redox reactions because of the high surface area of the V_2_O_5_ nanobelts anchored to the rGO layers. However, excessive V_2_O_5_ content could lead to agglomeration that decreases the performance of the VrG composite. Fu et al. found a balance between the rGO and V_2_O_5_ contents by synthesizing VrG electrodes with different weight percentages of V_2_O_5_ [74]. The fabricated VrG composite synthesized via microwave synthesis method resulted in a uniform distribution of amorphous V_2_O_5_ nanoparticles on lamellar rGO sheets. Flaky rGO prevented the agglomeration of V_2_O_5_, which is a common problem that decreases the surface ion adsorption capability of the V_2_O_5_ nanostructure. More V_2_O_5_ crystals were formed on the rGO surface without agglomeration at a high initial V_2_O_5_ loading up to 34.1 wt%. VrG-34 (34.1 wt% V_2_O_5_) exhibited excellent electrochemical performance with a C_sp_ of 673.2 F g^−1^ at a current density of 1 A g^−1^. The high rGO content resulted in high rate capability, as indicated by a large C_sp_ of 474.6 F g^−1^ at a high current density of 10 A g^−1^. A high V_2_O_5_ composition decreased the ion adsorption and number of interlayer rGO bridges, which are essential for fast electron transport. With an increase in current density, the high capacitance shifted from the redox dependence of V_2_O_5_ to the capacitive dependence provided by rGO. The porous and stable architecture of the composite allowed good cycling with a 96.8% C_sp_ retention after 10,000 cycles at a current density of 1 A g^−1^. A symmetric supercapacitor exhibited good performance with an energy density of 46.8 Wh kg^−1^ at a power density of 499.4 W kg^−1^. A high V_2_O_5_ content is necessary for high energy density through faradic energy storage; but ultimately, a balance between V_2_O_5_ and rGO is essential for maximizing the electrochemical performance.

A variation in the ratio of rGO to V_2_O_5_ also alters the porosity and surface area of the final VrG composite. Typically, a greater amount of rGO results in a larger surface area owing to its high surface area. Yao et al. found that decreasing V_2_O_5_ content increased the porosity of the composite [75]. A mixture of V_2_O_5_ powder with varying initial loading, H_2_O_2_, and GO sols was heated at 200 °C for 5 h to yield a VrG composite. The resulting VrG aerogel comprised V_2_O_5_ nanobelts with a pure orthorhombic crystal phase uniformly intercalated between the layers of rGO. VrG with 80.5 wt% V_2_O_5_ exhibited a small BET surface area of 20.5 m^2^ g^−1^, whereas VrG with 15.6 wt% V_2_O_5_ exhibited a large BET surface area of 103.4 m^2^ g^−1^. As rGO had a greater surface area than V_2_O_5_, the addition of higher amounts of rGO increased the overall surface area of the VrG material. Moreover, V_2_O_5_ tended to agglomerate at high concentrations. A decrease in the total V_2_O_5_ content resulted in a more homogeneous distribution of V_2_O_5_ throughout the rGO surface to provide more active sites. VrG-62 (61.6 wt% V_2_O_5_) outperformed other VrG electrodes owing to a good balance of EDLC and pseudocapacitance while maintaining a high pore volume of 0.008 cm^3^ g^−1^. The VrG-62 electrode exhibited a maximum C_sp_ of 310.1 F g^−1^ at a current density of 1 A g^−1^. The VrG composite had good rate capability, as indicated by a high C_sp_ of 195.2 F g^−1^ at a large current density of 10 A g^−1^. Choudhury et al. also reported that a lower V_2_O_5_ content of a V_2_O_5_/graphene (VG) composite increased the specific surface area [76]. The VG material was synthesized via an in situ chemical reaction between V_2_O_5_ powder and exfoliated graphene with H_2_O_2_. V_4_G_1_ (4:1 V_2_O_5_ to graphene) and V_2_G_1_ (2:1 V_2_O_5_ to graphene) exhibited a layered structure with V_2_O_5_ nanofibers anchored between the layers of rough graphene. The BET surface area of V_2_G_1_ was 142 m^2^ g^−1^, which decreased to 117 m^2^ g^−1^ for V_4_G_1_. A low amount of V_2_O_5_ produced a composite with a high surface area, resulting in more active redox sites and greater intercalation/de-intercalation. Accordingly, V_2_G_1_ exhibited a higher C_sp_ of 218 F g^−1^ compared to 112 F g^−1^ for V_4_G_1_ at a current density of 1 A g^−1^. A symmetric supercapacitor of V_2_G_1_ yielded an energy density of 22 Wh kg^−1^ at a power density of 3594 W kg^−1^.

However, a balance between V_2_O_5_ and rGO could increase the surface area to greater than that of pure rGO. Deng et al. found that a VrG composite yielded higher porosity than that of pure rGO because of the increased interlayer spacing of the rGO sheets caused by V_2_O_5_ intercalation [77]. A VrG monolith was synthesized via a hydrothermal method, resulting in a lamellar rGO structure with V_2_O_5_ nanowires embedded between the thin rGO sheets (22 wt% rGO). The resulting BET surface area of the VrG monolith was 172.9 m^2^ g^−1^, which was ~100 m^2^ g^−1^ larger than that of a pure rGO monolith and approximately five times larger than that of pure V_2_O_5_. The VrG monolith was also compressed into a thin film electrode. The C_sp_ of VrG monolith and thin-film were 385 and 272 F g^−1^, respectively, at a current density of 0.25 A g^−1^. The VrG material exhibited enhanced ion diffusion rates owing to increased porosity, resulting in good rate capability, as indicated by a high C_sp_ of 224 F g^−1^ at a current density of 10 A g^−1^. The energy density of the asymmetric supercapacitor with the VrG electrode was 26.22 Wh kg^−1^ at a power density of 425 W kg^−1^, which decreased to 7 Wh kg^−1^ at a power density of 8500 W kg^−1^. Ndiaye found that an increase in the graphene foam (GF) content only increased the porosity to a certain extent [78]. A V_2_O_5_/GF composite was synthesized with varying initial GF loading. The BET surface area increased from 4.9 m^2^ g^−1^ for pristine V_2_O_5_ nanosheets to 5.1 m^2^ g^−1^ when 50 mg of GF was added. Because GF has a high specific surface area of 208.8 m^2^ g^−1^, the initial integration of GF increased the overall surface area. The low GF content also prevented the homogeneous integration of the V_2_O_5_ nanosheets with GF. An increase in the GF loading to 150 mg (VrG-150) resulted in a maximum surface area of 9.5 m^2^ g^−1^. However, further increase in the GF loading to 200 mg resulted in the reduction of surface area to 6.2 m^2^ g^−1^. A high GF content caused GF and V_2_O_5_ to form separate agglomerations instead of a homogeneous composite. Consequently, the porous and homogeneous VrG-150 outperformed the other electrodes, with a specific capacity of 73 mAh g^−1^ at a current density of 1 A g^−1^. An asymmetric capacitor assembled using a VrG-150 positive electrode and carbonized Fe-adsorbed polyaniline (C-FE-PANI) negative electrode yielded a specific capacity of 41 mAh g^−1^ at a current density of 1 A g^−1^. The energy density was 39 W h kg^−1^ at a power density of 947 W kg^−1^.

Determining the optimal ratio between rGO and V_2_O_5_ is essential for synthesizing a high-performance electrode material with a balance between the EDL and pseudocapacitive characteristics. An appropriate rGO content improves the conductivity and electrode cyclability, whereas a sufficient V_2_O_5_ content is necessary for high specific capacitance. The addition of rGO to bulk V_2_O_5_ increases the surface area of the composite material, and the intercalation of V_2_O_5_ into rGO layers can also increase the porosity of the composite material. However, a balance between the two components is necessary to prevent agglomeration.

### 2.3. Effects of Physical Treatment

Annealing newly fabricated VrG composites can alter oxidation states of vanadium oxides and improve their crystallinity, resulting in enhanced electrochemical performance. Control of the temperature during the heat treatment can increase the oxidation of V_2_O_5_ and the formation of a homogeneous composite. Following a hydrothermal synthesis procedure, Li et al. synthesized V_x_O_y_ nanoflowers anchored to rGO sheets [79]. Annealing of the resulting composite caused the nanoflowers to morph into V_2_O_5_ nanorods with varying lengths and diameters depending on the annealing temperature. Moreover, an increase in the annealing temperature converted amorphous vanadium oxides into orthorhombic V_2_O_5_ crystals through the oxidation of V^4+^ to V^5+^. VrG annealed at 350 °C under nitrogen atmosphere exhibited the maximum C_sp_ of 537 F g^−1^ at a current density of 1 A g^−1^ due to short ion diffusion pathways in the organized V_2_O_5_ crystal lattice. Because the nanorods were uniformly anchored to the mesoporous rGO surface, the high conductivity and porosity afforded by the rGO resulted in a good rate capability, indicated by the 60% C_sp_ retention at a high current density of 20 A g^−1^. Improved cyclic performance of the VrG material was also observed, with 84% C_sp_ retention after 1000 cycles at a current density of 1 A g^−1^. The VrG electrode exhibited a high energy density of 74.58 Wh kg^−1^ at a power density of 500 W kg^−1^, which decreased slightly to 29.33 Wh kg^−1^ at a power density of 10,000 W kg^−1^. In a similar study, Liu et al. fabricated a VrG composite via a hydrothermal method using an NH_4_VO_3_ precursor followed by annealing at different temperatures in air [80]. Without annealing, V_2_O_5_ nanospheres of inconsistent sizes and incomplete formation were observed, whereas VrG annealed at a high temperature of 350 °C (VrG-350) exhibited a degraded rGO support without distinct V_2_O_5_ nanospheres. However, VrG annealed at 300 °C (VrG-300) had uniformly sized V_2_O_5_ nanospheres homogeneously embedded on the surface of the rGO. VrG-300 exhibited greater GO reduction and an increase in V_2_O_5_ composition from 12 to 30 wt%, owing to greater V^4+^ oxidation to V^5+^. With fewer defects in the carbon lattice and a higher V_2_O_5_ content, VrG-300 exhibited the largest C_sp_ of 386 F g^−1^ at a current density of 0.5 A g^−1^. An asymmetric supercapacitor with the VrG-300 electrode had a maximum energy density of 80.4 Wh kg^−1^ at a power density of 275 W kg^−1^, which decreased to 32 Wh kg^−1^ at a power density of 1374 W kg^−1^. Thangappan et al. electro-spun a mixture of vanadium acetylacetonate, GO, polyvinyl pyrrolidone, dimethylformamide (DMF), and ethanol, and subsequently annealed the composite at 350 or 550 °C in air [81]. Non-annealed VrG and VrG annealed at 350 °C resulted in a web of uniformly thin and straight nanowires. VrG annealed at 550 °C resulted in a twisted and aggregated structure with arched webs that were half the diameter of the non-annealed VrG due to the decomposition of the organic binder material. Annealing at 550 °C completely reduced GO, resulting in a more conductive composite fiber. However, high temperature disintegrated the carbon material, resulting in a low carbon content of 0.3 wt%. Non-annealed and low-temperature annealed VrG had amorphous crystal structures, whereas the VrG annealed at 550 °C possessed an orthorhombic crystal structure, as indicated by the formation of large V_2_O_5_ crystals. The stable crystals and mesoporous, web-like structure of VrG annealed at 550 °C allowed a large amount of ion diffusion into the electrode for high energy storage. The C_sp_ was 453.8 F g^−1^ at a scan rate of 10 mV s^−1^ because of the slow ion intercalation at low scan rates. The C_sp_ decreased to 111.01 F g^−1^ at a scan rate of 100 mV s^−1^. Therefore, the control of the annealing temperature is essential for both the reduction of GO to rGO and improving the crystallinity of V_2_O_5_.

Despite the previously mentioned benefits of heat treatment, the annealing of VrG electrodes may decrease the electrical performance. Lee et al. analyzed the discrepancy in the electrochemical performance of a non-annealed V_2_O_5_ composite in comparison with an annealed composite, where an rGO thin film was fabricated from GO using CO_2_ laser reduction, and V_2_O_5_ was deposited on the film via atomic layer deposition [82]. The CO_2_ laser changed the porosity of the compact GO by increasing the interstitial gap between the rGO sheets while creating pores that penetrated multiple rGO sheets. The spacing between the rGO sheets increased with V_2_O_5_ infiltration. The V_2_O_5_ deposition left the rGO template intact, and further annealing the composite in an argon atmosphere did not significantly change the surface morphology of the electrode. However, annealing converted the amorphous V_2_O_5_ crystal structure into a more crystalline V_2_O_5_ with varying oxidation states of vanadium oxide. The amorphous crystal characteristics of V_2_O_5_ before annealing allowed deeper ion diffusion through the distorted lattices, resulting in a greater faradic character. In contrast, the annealed VrG exhibited higher crystallinity and greater EDL contribution. Considering the lamellar structure of the rGO sheets, the diffusion-based faradic capacitance of the non-annealed VrG resulted in a higher C_sp_ of 189 F g^−1^ at a current density of 1 A g^−1^. The charge transfer resistance was also lower for the non-annealed VrG. Initial cycling increased the C_sp_ to 108% for the non-annealed VrG material owing to electro-activation caused by the initial intercalation of electrolyte ions. However, this did not occur for annealed VrG because of the more rigid crystallized structure of the annealed VrG (Figure 4).

In addition to heat treatment, the morphology and crystallinity of a VrG composite can be altered by laser irradiation. Lazauskas et al. first synthesized a V_2_O_5_/GO nanoribbon composite using a melt-quenching process and further applied laser treatment to reduce GO to rGO [83]. A 405-nm laser was used to irradiate specific regions of the V_2_O_5_/GO composite, resulting in a VrG composite with surface protrusions. The laser treatment resulted in pillar-like V_2_O_5_ protrusions that uniformly extruded from the rGO base. These protrusions increased the surface area by four times to 17.27 m^2^ g^−1^. An increase in the laser power output from 1.69 to 2.03 W cm^−2^ to 2.37 W cm^−2^ resulted in more protrusions. However, further increase in power to 2.71 W cm^−2^ led to larger V_2_O_5_ agglomerations, thereby decreasing the surface area of the composite material. The laser treatment also decreased the amount of intercalated H_2_O molecules in the V_2_O_5_ crystal lattice while further reducing GO to rGO. The laser treatment resulted in a VrG composite with a more organized V_2_O_5_ crystal structure and high conductivity of 6.8 S m^−1^.

Physical treatment of V_2_O_5_ with heat or laser can alter the crystallinity of V_2_O_5_, oxidize V^4+^ to V^5+^ and reduce GO to rGO. Highly crystalline V_2_O_5_ is more conductive than amorphous V_2_O_5_, allowing for greater rate capability, but it is less efficient for ion diffusion than the latter. The reduction of GO to rGO improves the conductivity of the carbon backbone in the VrG composite. Physical treatment can also be used to control the formation of agglomerations, and excessive heat exposure can destroy the carbon content of the hybrid material.

## 3. V_2_O_5_/CNT

CNTs have received considerable attention owing to their high conductivities, large surface areas, mechanical stability, and unique 1D tubular structures [84,85]. The ability of CNTs to be functionalized allows precise control of their properties to meet the needs of specific supercapacitor applications [86]. Single-walled CNTs (SWCNTs) and multi-walled CNTs (MWCNTs) have been utilized in conjunction with vanadium oxides to yield high-performance hybrid materials for supercapacitor applications [87]. Herein, the effects of different synthesis pathways, V_2_O_5_/CNT compositions, and physical treatments on the morphology and electrochemical performances of V_2_O_5_/CNT (VCNT) composite materials are discussed. The morphology and electrochemical performances of V_2_O_5_/CNT electrodes for supercapacitor applications reported in the literature are summarized in Table 2.

### 3.1. Effects of Synthesis Method

A common procedure for fabricating VCNT composites is the hydro/solvothermal method. Jiang et al. grew V_2_O_5_ crystals on a GF/CNT substrate by employing a hydrothermal process [88]. Subsequently, PEDOT coating was deposited via a chronoamperometry technique, yielding a VCNT/GF/PEDOT composite (Figure 5).

Orthorhombic crystal phase V_2_O_5_ nanostars were intercalated into the interstitial space between the vertically aligned CNTs. The addition of the PEDOT coating reduced the V_2_O_5_ dissolution during synthesis and increased the size of the V_2_O_5_ nanostars by ~15%. The high surface area of the enlarged V_2_O_5_ nanostars increased the pseudocapacitive surface reactions. The composite electrode was tested in a three-electrode configuration with a Pt counter electrode and an Ag/AgCl reference electrode in 5 M LiNO_3_ electrolyte, yielding a high C_sp_ of 1016 F g^−1^ at a current density of 1 A g^−1^. The direct growth of V_2_O_5_ on the vertically aligned CNTs enabled good surface contact for efficient electrical conduction. The porous Ni foam/CNT frame also provided structural stability to the overall electrode. The result indicated good rate capability with an excellent C_sp_ of 484 F g^−1^ at a high current density of 20 A g^−1^ and high coulombic efficiency of 97.8%. The asymmetric supercapacitor with a VCNT electrode exhibited a maximum energy density of 13.24 Wh kg^−1^ at a power density of 710 W kg^−1^, which decreased to 10.34 Wh kg^−1^ at a power density of 2659 W kg^−1^. The energy density decreased slightly at high power owing to the absence of an insulating binder material and the high conductivity provided by the Ni/CNT frame. Wang et al. synthesized a VCNT composite by adding CNTs to the V_2_O_5_ nanosheets that were synthesized via a hydrothermal reaction between bulk V_2_O_5_ and H_2_O [89]. CNTs were subsequently added to a solution of V_2_O_5_ nanosheets and aged for two days. This fabrication method resulted in a lamellar nanostructure with V_2_O_5_ layers intercalated with CNTs, corresponding to 13.7 wt% of the total composition. The CNTs increased the interstitial spacing between the V_2_O_5_ nanosheets, preventing V_2_O_5_ agglomeration and increasing the porosity, yielding a large specific surface area of 102.05 m^2^ g^−1^. The C_sp_ of the VCNTs was 553.33 F g^−1^ at a current density of 5 mA cm^−2^. The increased interlayer spacing between the V_2_O_5_ planes allowed greater ion intercalation/de-intercalation, which promoted faradic redox reactions. Furthermore, the CNTs increased the overall conductivity of the device, resulting in a low series resistance of 0.95 Ω and a charge transfer resistance of 0.6 Ω for enhanced rate capability. Owing to the structural stability provided by the CNTs, the charge/discharge stability increased, as indicated by a high C_sp_ retention of 83% after 1000 cycles at a current density of 10 mA cm^−2^.

Deposition techniques have also been used to grow vertically aligned CNTs and V_2_O_5_ composites directly on conductive substrates such as Ni, yielding a binderless electrode material. Jampani et al. used CVD to synthesize an array of vertically aligned CNTs on flat Ni disks that formed a forest-like microstructure. Atmospheric pressure CVD was used to deposit Ti-doped V_2_O_5_ on the CNTs [90]. The resulting composite had an amorphous crystal phase with V_2_O_5_ globules anchored to the vertically aligned CNT forest; a C_sp_ of 313 F g^−1^ was observed at a scan rate of 2 mV s^−1^. The amorphous crystal structure of the Ti-doped V_2_O_5_ decreased the V_2_O_5_ dissolution during charge/discharge, resulting in nearly perfect C_sp_ retention even after 400 cycles at a high scan rate of 100 mV s^−1^. The areal capacitance was 350 mF cm^−2^ at a scan rate of 2 mV s^−1^, which decreased to 20 mF cm^−2^ at a scan rate of 200 mV s^−1^. The decent rate capability was due to the low resistance of the binderless composite at 0.0182 Ω cm ^2^, which was two orders of magnitude more conductive than pristine V_2_O_5_. The electrodes had a maximum energy density of 25 Wh kg^−1^ at a power density of 100 W kg^−1^, which decreased to 11 Wh kg^−1^ at a power density of 4500 W kg^−1^. Shakir et al. used spray coating to prepare a layered V_2_O_5_/MWCNT (VMWCNT) composite electrode on a Ni/Cu/Ni/Au electroplated substrate [91]. A polyester fiber fabric was first plated with Ni/Cu/Ni/Au layers sequentially via electroless plating. The MWCNT/V_2_O_5_ core/shell nanotubes were synthesized via a bottom-up assembly method using pre-functionalized MWCNTs and NH_4_VO_3_. A layer of graphene was then deposited on the conductive fabric. Subsequently, a mixture of V_2_O_5_-coated MWCNTs was spray-coated on the top graphene layer. Alternating graphene and VMWCNT layers were applied repeatedly to yield a binderless VMWCNT/graphene composite electrode. The electrode comprised a thick orthorhombic V_2_O_5_ shell coating the MWCNTs. The C_sp_ of the VMWCNT/graphene electrode with 3-nm-thick V_2_O_5_ layers had an excellent C_sp_ of 2590 F g^−1^ at a scan rate of 1 mV s^−1^. Increasing the V_2_O_5_ coating thickness to 20 nm decreased the C_sp_ to 510 F g^−1^ because of inefficient ion penetration into the inner layers of the VMWCNTs. The 3-nm-thick VMWCNT/graphene electrode retained 96% of its initial C_sp_ after 5000 cycles at a scan rate of 20 mV s^−1^ because of the effective conductive MWCNT bridge linking electron transport from V_2_O_5_ to graphene. The energy density was 96 Wh kg^−1^ at a power density of 800 W kg^−1^, which decreased to 28 Wh kg^−1^ at a power density of 9000 W kg^−1^.

One major advantage of CNTs is their easy functionalization to yield high-performance VCNT electrodes through various synthesis methods. Modifying CNTs with specific functional groups such as hydroxyl or carboxyl groups can improve the conductivity of the CNTs and promote the nucleation of V_2_O_5_ crystals. Hu et al. used a one-step hydrothermal method to synthesize a vanadium oxide composite with functionalized CNTs [92]. The CNTs were functionalized with hydroxyl and carbonyl groups using concentrated sulfuric and nitric acids, which acted as the activation centers for coordinate bond formation with VO. Vanadium oxide nanoribbons were formed at the nucleation sites via a hydrothermal reaction and were bonded to the CNTs via hydroxyl functional groups. By adding GO to a mixture of V_2_O_5_ and CNTs, Hu et al. fabricated a V_3_O_7_/CNT/rGO composite (rGO-VCNT) comprised of rGO sheets uniformly coated with large V_3_O_7_ nanobelts and thin CNTs. The rGO-VCNT composite with 40 wt% rGO exhibited a maximum C_sp_ of 685 F g^−1^ at a current density of 0.5 A g^−1^, which decreased to ~375 F g^−1^ at a high current density of 10 A g^−1^. The high porosity of rGO and improved conductivity because of well-distributed CNTs resulted in an excellent rate capability. With a primarily carbon-based composition and strong bonds between V_2_O_5_ and CNTs, the rGO-VCNT (40 wt% rGO) electrode retained 99.7% of its initial C_sp_ after 10,000 cycles at a high scan rate of 100 mV s^−1^. The maximum energy density of the symmetric supercapacitor was 34.3 Wh kg^−1^ at a power density of 150 W kg^−1^, which decreased to 18.8 Wh kg^−1^ at a power density of 3000 W kg^−1^. Mtz-Enriquez et al. reported an improved discharge time of a VCNT electrode by functionalizing the CNTs to promote V_2_O_5_ defects [93]. The CNTs were first deposited on a flexible graphene electrode and activated with strong acids. A V_2_O_5_ slurry was then applied to the CNT-coated graphene electrode and hot-pressed at 0.1 ton for physical binding. The resulting composite was comprised of rectangular V_2_O_5_ orthorhombic nanoribbons intercalated into a fibrous CNT structure. Oxygen vacancies formed in the V_2_O_5_ crystal structure due to the reduction of V^5+^ to V^4+^_,_ which acted as redox centers for delayed current discharge. The functionalization of CNTs with carboxylic groups also created oxygen vacancies in the carbon lattice, promoting the formation of additional V_2_O_5_ defects. The CNTs also provided a secondary layer for ion storage and a large surface area for V_2_O_5_ contact, resulting in a high capacity and rate capability. The asymmetric supercapacitor with a graphene/VCNT anode had an energy density of 369.6 Wh kg^−1^.

MWCNTs can also be functionalized to achieve an increase in conductivity and homogeneous V_2_O_5_ formation. Saravanakumar et al. functionalized MWCNTs with hydroxyl, carboxyl, and keto-carbonyl groups using concentrated H_2_SO_4_ and HNO_3_ [94]. A mixture of functionalized MWCNTs and V_2_O_5_ crystals was aged for three days at room temperature to yield a V_2_O_5_/MWCNT (VMWCNT) composite. The orthorhombic V_2_O_5_ crystals formed the cores surrounded by MWCNTs, resulting in a highly porous network. Even with a relatively low carbon content of 8.73 wt%, the surface area was large (14.4 m^2^ g^−1^) because of the mesoporous web-like architecture of the MWCNTs. The VMWCNT electrode exhibited a maximum C_sp_ of 410 F g^−1^ at a current density of 0.5 A g^−1^, which decreased to 280 F g^−1^ at a current density of 10 A g^−1^. The high rate capability was because of the porous structure that allowed the fast intercalation/de-intercalation of ions and high conductivity of the functionalized CNT web. The functionalized CNTs improved the cyclability of the VMWCNT electrode, as indicated by 86% C_sp_ retention after 600 cycles at a high current density of 10 A g^−1^. The symmetric supercapacitor with functionalized VMWCNT electrodes exhibited an energy density of 8.9 Wh kg^−1^ at a power density of 121 W kg^−1^. Pandit et al. functionalized MWCNTs with carboxyl groups using H_2_O_2_ [95]. A stainless-steel current collector was first dip coated with the MWCNTs, and the resulting MWCNT/stainless-steel electrode was dip coated with a solution of VOSO_4_ and NaOH. After repeated dipping and drying, the resulting VMWCNT/stainless-steel electrode had a layered structure of orthorhombic V_2_O_5_ flakes intercalated in a web of long MWCNTs. A C_sp_ of 629 F g^−1^ was observed at a current density of 2 A g^−1^, which decreased to ~320 F g^−1^ at a current density of 8 A g^−1^. The C_sp_ of the composite electrode was high because the porous structure formed by the web-like MWCNTs allowed large volumes of counterion intercalation/de-intercalation deep in the V_2_O_5_ crystals. The VMWCNT electrode retained 93% of its C_sp_ after 4000 cycles at a high scan rate of 100 mV s^−1^, showing good reversibility and stability. A symmetric supercapacitor with VMWCNT electrodes yielded an energy density of 72 Wh kg^−1^ at a power density of 2300 W kg^−1^ that decreased to 18.66 Wh kg^−1^ at a power density of 8400 W kg^−1^ and current density of 4 A g^−1^.

Different synthesis strategies, including hydro/solvothermal and deposition methods, can result in VCNT composites with different nanostructures. The hydrothermal method is a facile technique that involves the nucleation of V_2_O_5_ crystals from vanadium precursors on CNTs at high temperatures. Deposition techniques such as CVD allow the direct layering of V_2_O_5_ on CNTs with a controllable thickness. Functionalizing the CNTs with hydroxyl and carbonyl groups before synthesis with V_2_O_5_ can improve the overall conductivity, porosity, and stability of the hybrid material.

### 3.2. Effects of Composition

The ratio of V_2_O_5_ to CNTs affects the electrochemical performance of VCNT electrodes via changes in their morphologies, conductivities, porosities, and pseudocapacitive potentials. Guo et al. reported increases in the conductivity and rate capability with high CNT content [96]. The V_2_O_5_ᐧ*n*H_2_O aerogel and CNTs were dispersed in DI water, vacuum filtered with a cellulose membrane, and dried at 60 °C. The film was comprised of densely packed V_2_O_5_ nanosheets with homogeneously intercalated CNTs between the V_2_O_5_ layers, preventing V_2_O_5_ self-stacking. An increase in the CNT content increased the thickness of the VCNT film by increasing the interstitial space between the V_2_O_5_ sheets. The increased thickness prevented deep ion infiltration into the inner V_2_O_5_ layers near the center of the electrode. However, an increase in the CNT content increased the conductivity of the overall electrode by introducing more interlayer CNT connections, as indicated by the low square resistance of 128 Ω sq^−1^ for VCNT-10 (10 wt% CNT). VCNT-10 exhibited the highest C_sp_ of 207.7 F g^−1^ (521 F cm^−3^) at a current density of 0.5 A g^−1^. An increase in the CNT content increased the rate capability, with VCNT-15 (15 wt% CNT) exhibiting the highest C_sp_ retention of 49.5% compared to the 42% retention of the VCNT-10 electrode at a high current density of 20 A g^−1^. The flexible symmetric supercapacitor exhibited good performance in both bent and unbent states with a high energy density of 7300 Wh kg^−1^ at a power density of 42.6 W kg^−1^. Yilmaz et al. reported similar improvements with fewer V_2_O_5_ nanosheets intercalated with CNT [97]. V_2_O_5_ gel was rigorously mixed in solutions with different amounts of CNTs, yielding free-standing electrodes consisting of V_2_O_5_ nanosheets with an orthorhombic crystal structure. The CNTs were homogeneously intercalated between the V_2_O_5_ nanosheets. An increase in the V_2_O_5_ content decreased the porosity of the V_2_O_5_ nanosheets, decreasing the overall surface area of the VCNT composite. The composite gel was printed on a 1 cm × 1 cm ITO glass plate and dried before use as an electrode. V_1_CNT_2_ (1:2 ratio of V_2_O_5_ to CNT) yielded the highest C_sp_ of 116 F g^−1^ at a current density of 0.1 A g^−1^. An increase in the V_2_O_5_ loading to 1:1 ratio (V_1_CNT_1_) resulted in a lower C_sp_ value of 32 F g^−1^ at a current density of 0.1 A g^−1^ because of a decrease in the number of active sites on V_2_O_5_ nanosheets. Further addition of V_2_O_5_ decreased the C_sp_ due to V_2_O_5_ agglomeration, which decreased surface ion adsorption. Because of the highly conductive CNT linkages, the carbon-dominant V_1_CNT_2_ electrode exhibited a good cycling capability, with 91.2% capacitance retention after 5000 cycles at a current density of 5 A g^−1^. A symmetric supercapacitor with V_1_CNT_2_ electrode yielded a volumetric energy density of 0.67 mWh cm^−3^ at a power density of 0.27 W cm^−3^. Wu et al. fabricated a thin-film electrode with varying V_2_O_5_ content using a sol-gel method [98]. V_2_O_5_ nanobelts, CNTs, ethanol, terpilneol, and ethyl cellulose were mixed at 80 °C to yield a gel that was then doctor bladed onto a ceramic plate and annealed at 350 °C in air. The thin film was peeled off the ceramic to yield a free-standing VCNT thin film. The V_2_O_5_ nanobelts were highly uniform with an orthorhombic crystal phase, resulting in uniformly thick (20–100 nm) and long (>100 nm) nanobelts. A conductive current collector was added by compressing the thin-film electrode between Ti foils at 4000 psi. The VCNT-75 (75 wt% V_2_O_5_ composition) thin-film exhibited the highest C_sp_ of 216 F g^−1^ at a scan rate of 5 mV s^−1^, which was five times greater than that of the bare CNTs. VCNT-75 also exhibited the highest volumetric capacitance of 540 F cm^−3^ at a scan rate of 5 mV s^−1^, which decreased by 79% at a scan rate of 100 mV s^−1^. The poor rate capability was due to an inefficient electron transfer caused by the low carbon composition. VCNT-61 (60.5 wt% V_2_O_5_) exhibited the best balance between maximum capacitance and electrode stability with a C_sp_ of 192 F g^−1^ at a scan rate of 5 mV s^−1^ and a 64% C_sp_ retention at a scan rate of 100 mV s^−1^. Moreover, the higher CNT content increased the electron conduction, resulting in 79.8% C_sp_ retention after 5000 cycles at a scan rate of 50 mV s^−1^. A symmetric supercapacitor with VCNT-61 electrodes yielded a maximum volumetric energy density of 41 Wh L^−1^ at a volumetric power density of ~400 W L^−1^, which decreased to 29.1 Wh L^−1^ at a volumetric power density of 6500 W L^−1^. A balance between conductive CNTs and energy-dense V_2_O_5_ is necessary for achieving the maximum electrochemical performance.

The CNT to V_2_O_5_ ratio changes the extent of the EDL and pseudocapacitive characteristics of the hybrid material. Perera et al. found that altering the V_2_O_5_ to CNT ratio can also affect the physical properties of the composite thin film [99]. A free-standing VCNT electrode was fabricated by filtering a suspension of CNTs and V_2_O_5_ through nylon filter paper. The VCNT-covered filter papers were dried, yielding a highly flexible thin-film supercapacitor electrode. The hybrid material was comprised of long V_2_O_5_ nanowires that formed a web-like structure. Short and curly CNTs were uniformly intercalated into the V_2_O_5_ web. Increasing V_2_O_5_ content decreased the porosity of the composite while increasing the CNT content increased the brittleness of the thin-film electrode. A 1:1 ratio of V_2_O_5_ and CNTs (V_1_CNT_1_) exhibited the greatest structural integrity and optimal electrochemical performance with a C_sp_ of 57.3 F g^−1^ at a current density of 0.5 A g^−1^. The V_1_CNT_1_ electrode exhibited good rate capability with a C_sp_ of 42.9 F g^−1^ at a current density of 10 A g^−1^. A coin-cell type supercapacitor with a V_1_CNT_1_ anode exhibited an energy density of 46.3 Wh kg^−1^ at a power density of 5260 W kg^−1^. In comparison, the VCNTs with a 1:5 ratio of V_2_O_5_ to CNTs had a lower energy density of 6 Wh kg^−1^ because of the decreased pseudocapacitive energy storage capacity. A good balance of V_2_O_5_ content for maximum pseudocapacitive capacity and increased CNT content for improved porosity is essential for optimized VCNT electrode performance. Such flexible, high-performance electrodes allow the development of more powerful portable and bendable technologies. Sathiya et al. reported different contributions of intercalative and capacitive charge capacities by varying the V_2_O_5_ content [100]. VTIP was dispersed in a suspension of pre-functionalized CNTs and stirred for one day to allow slow hydrolysis. The resulting VCNT composite gel was aged for a week, washed with acetone, dried at room temperature, and further dried at 200 °C. Long CNTs were uniformly coated with a layer of V_2_O_5_ crystals, resulting in a web-like structure of long V_2_O_5_/CNT core/shell nanotubes. Pristine CNTs relied purely on nonfaradic mechanisms for charge storage, whereas pure V_2_O_5_ mainly used faradic ion intercalation for charge storage. An increase in the V_2_O_5_ content from 0 to 25 wt% increased both faradic and nonfaradic contributions owing to the synergistic effect between the CNTs and V_2_O_5_. However, further increase in the V_2_O_5_ content decreased the faradic contribution due to increased V_2_O_5_ coating thickness. With a V_2_O_5_ loading of 25 wt%, 67% of the total stored charge was from pseudocapacitive redox reactions and surface ion adsorption. The maximum specific capacity was 850 mAh g^−1^, which was more than twice the capacity of purely pseudocapacitive pristine V_2_O_5_.

Because essential factors, including conductivity, porosity, and the charge storage method, are dependent on the ratio of V_2_O_5_ to CNT, it is important to optimize each type of VCNT composite. In general, a higher CNT content increases the EDL contribution, conductivity, and rate capability. In contrast, a large V_2_O_5_ content results in a higher faradic contribution, energy storage potential, and flexibility.

### 3.3. Effects of Physical Treatment

Heat treatment is often used as the final step in VCNT synthesis because annealing the composite material improves the crystallinity and morphology of the composite. Sun et al. found an increased conversion of amorphous vanadium oxide to V_2_O_5_ after heat treatment [101]. Vertically aligned CNTs were deposited on a Si substrate via CVD and reacted with vanadium(III) acetylacetonate. Upon annealing at 350 °C in air, V_2_O_5_ microspheres were uniformly grown on the CNT surface. Annealing the composite converted the vanadium precursor to V_2_O_5_ and improved the crystallinity of the vanadium oxide precursor. Even after high-temperature annealing, the composite contained mixed valence states of vanadium oxide, which enhanced the ion diffusion. The C_sp_ of the VCNT composite was 284 F g^−1^ at a current density of 2 A g^−1^, which decreased by ~50% at a high current density of 15 A g^−1^. The direct growth of V_2_O_5_ on the CNTs improved the surface contact, thereby decreasing the charge transfer resistance to 1.2 Ω and resulting in good rate performance. Moreover, interstitial spacing between the vertically aligned CNTs provided a spacious environment that decreased the internal strain during charge/discharge, resulting in a high C_sp_ retention of 89.3% after 2500 cycles at a high current density of 10 A g^−1^. An asymmetric supercapacitor with the VCNT electrode had an energy density of 32.3 Wh kg^−1^ at a power density of 118 W kg^−1^. Similarly, Wang et al. reported increased V_2_O_5_ formation after calcination [102]. A solution of NH_4_VO_3_, CNTs, super AC (SAC), and DI water was reacted in an autoclave at 180 °C for one day and subsequently calcinated in a muffle furnace at 250 °C for 6 h in air. The resulting VCNT material had orthorhombic V_2_O_5_ sheets interspaced with thick CNTs and SAC bundles. The addition of CNTs increased the BET surface area from 14 to 19 m^2^ g^−1^, and the addition of both CNTs and SAC further increased the surface area to 78 m^2^ g^−1^ because of the increased mesopore volume. The VCNT electrode without SAC outperformed pristine V_2_O_5_ with a C_sp_ of 231.8 F g^−1^ at a high current density of 10 A g^−1^. A high C_sp_ was observed even at a high current density because of the improved conductivity provided by the intercalated CNTs and short ion diffusion pathways to the crystalline V_2_O_5_ sheets. Moreover, the VCNT electrode with SAC yielded a maximum C_sp_ of 357.5 F g^−1^ because of the increased surface area that provided more activation sites. Shakir et al. synthesized a VMWCNT material through a heat-induced reaction and further annealed the composite at 350 °C for 2 h under ambient conditions [103]. The resulting VMWCNT thin film had layers of highly orthorhombic V_2_O_5_ crystals surrounded by long and curly MWCNTs. VMWCNTs that were 3 nm thick exhibited the highest C_sp_ of 510 F g^−1^ at a scan rate of 1 mV s^−1^ because the thickness maximized the surface area-to-volume ratio of V_2_O_5_. The C_sp_ of the composite was significantly higher than that of bare MWCNTs at 80 F g^−1^ because of the pseudocapacitive reactions enabled by the V_2_O_5_ crystals. The VMWCNT electrode exhibited good stability with 96% of its initial C_sp_ retained after 5000 cycles at a high scan rate of 20 mV s^−1^ because of the improved conductivity of the MWCNTs and structural stability of the web-like nanostructure. The energy density of the electrode was 16 Wh kg^−1^ at a power density of 800 W kg^−1^. Physical treatment is a key component of VCNT synthesis because it promotes the conversion of amorphous vanadium oxides to a more crystalline V_2_O_5_.

## 4. V_2_O_5_/CNFs

CNFs have attracted significant attention as a substrate material for V_2_O_5_ growth because of their ease of synthesis, pore size controllability, and high conductivity [104]. Moreover, their stable 3D structures allow them to be used as free-standing electrodes for decreased internal resistance, high porosity, and flexible electrode applications [105]. V_2_O_5_/CNF (VCNF) composites often comprise either a CNF/V_2_O_5_ core/shell structure, V_2_O_5_ crystal growth on long CNFs, or V_2_O_5_ crystals intercalated in the CNF pores. The effects of the synthesis method, V_2_O_5_ content, and physical treatment on the electrochemical performances of the VCNF composite electrodes are reviewed. The morphology and electrochemical performances of V_2_O_5_/CNF electrodes for supercapacitor applications reported in the literature are summarized in Table 3.

### 4.1. Effects of Synthesis Method

Electrodeposition is a common technique used to form a V_2_O_5_ coating on long carbon fibers, resulting in a CNF/V_2_O_5_ core/shell structure. Song et al. used electrodeposition to coat the nanofibers of exfoliated carbon cloth (CC) with V_2_O_5_ [106]. The architecture of the fibrous structure was preserved, and two additional layers, an exfoliated carbon middle layer and a V_2_O_5_ outer layer, were added to the preexisting nanofibers. Due to a lack of heat or catalyst during electrodeposition, only half of the initial V_2_O_5_ precursor was oxidized, resulting in amorphous, mixed-valence V_2_O_5_. The areal capacitance of the device was 106 F cm^−2^ at a current density of 2 mA cm^−2^, which decreased by 24% at a current density of 20 mA cm^−2^. The good rate capability was attributed to the conductive carbon skeleton and improved electron mobility through the V-O-C bonds in the exfoliated carbon middle layer. Unlike most electrodes that degrade with repeated cycling, a 40% increase in the C_sp_ of the exfoliated VCNF electrode was observed after 10,000 cycles at a current density of 60 mA cm^−2^ because of the activation of the material via ion intercalation/de-intercalation. Repeated cycling introduced more intercalated water molecules, which expanded the interlayer distance for additional active sites. Water also shuttled the electrolyte ions during intercalation/de-intercalation. A similar V^4+^ to V^5+^ ratio decreased the dissolution of the V_2_O_5_ crystals during charge/discharge for improved stability.

Crystallization can be used to coat CNFs with V_2_O_5_. Zhou et al. synthesized a high-performance V_2_O_5_/polyindole/ACC (VPIAC) composite electrode via crystallization [107]. Activated carbon cloth (ACC) was synthesized via CC activation with HNO_3_, H_2_SO_4_, and KMnO_4_, followed by heat-induced reduction in N_2_/H_2_ (95%/5%) atmosphere at 1000 °C. Sodium metavanadate was dripped onto the ACC and crystallized at 400 °C. Subsequently, a solution of H_2_O_2_, indole, and DI water was dripped onto V_2_O_5_/ACC to promote V^4+^ oxidation to V^5+^. The resulting composite exhibited a bamboo-like structure with V_2_O_5_ and polyindole anchored to the surface of the long nanofibers. The electrochemical performance of the composite electrode was determined using a three-electrode configuration with a Pt counter electrode, Ag/AgCl reference electrode, and 5 M LiNO_3_ electrolyte. The VPIAC hybrid material exhibited a high C_sp_ of 535.3 F g^−1^ at a current density of 1 A g^−1^ because of the increased conductivity of the polyindole. The VPIAC electrode had a low series resistance and charge transfer resistance of 1.1 and 7.6 Ω, respectively. The hybrid material exhibited excellent 91.1% C_sp_ retention after 5000 cycles at a high current density of 10 A g^−1^ because the polyindole decreased the dissolution of V_2_O_5_ during charge/discharge and increased adhesion between the CNFs and V_2_O_5_. The energy density of an asymmetric supercapacitor with the VPIAC electrode was 38.7 Wh kg^−1^ at a power density of 900 W kg^−1^, which decreased to 32.6 Wh kg^−1^ at a power density of 18,000 W kg^−1^.

The sol-dipping method can also allow the uniform coating of CNFs with a V_2_O_5_ shell. Azadian et al. employed a sol-dipping method involving the repeated dipping and drying of polyacrylonitrile-based carbon paper into a V_2_O_5_ sol [108]. The dip–dry process was performed multiple times to promote the growth of a thick layer of V_2_O_5_ crystals on the carbon paper nanofibers. The V_2_O_5_ grown on the CNFs exhibited a high C_sp_ of 800 F g^−1^. Because of the porous architecture of the carbon fibers, the intercalation/de-intercalation was highly reversible, yielding a high Coulombic efficiency of 92%. The energy density of the electrode was 101 Wh kg^−1^ at a power density of 27 370 W kg^−1^.

In addition to the formation of a CNF/V_2_O_5_ core/shell nanostructure, small V_2_O_5_ nanoparticles can be intercalated within the porous CNFs. Zhou et al. fabricated a CC-based electrode with V_2_O_5_ nanosheets embedded within the CNF webs [109]. CC/graphene foam/CoMoS_4_ was mixed with a V_2_O_5_ supernatant for 12 h. The CoMoS_4_ bound to CC/graphene resulted in the uniform coating of thin CoMoS_4_ nanowires on the smooth CNFs, and the nanowires hosted the V_2_O_5_ nanosheets. The cyclic voltammetry (CV) curves of the hybrid material exhibited a distinct peak, indicating a large pseudocapacitive contribution. The CV curves retained their shape upon increasing scan rates from 5 to 50 mV s^−1^, indicating good rate capability owing to the conductive carbon matrix (Figure 6a). The composite behaved more like a battery instead of a capacitor, based on the large IR drop shown in the galvanostatic charge–discharge (GCD) plot in Figure 6b. The specific capacity of the electrode was 158.6 mAh g^−1^ at a current density of 1 A g^−1^ (Figure 6c).

Vanadium promoted faradic redox reactions, which increased the charge storage potential of the electrode. The addition of graphene to the carbon fibers enhanced the charge transfer from V_2_O_5_ to the current collector and functioned as a buffer to prevent physical deformation during intercalation/de-intercalation. Thus, the hybrid material was highly cyclable with a >90% C_sp_ retention after 5000 cycles (Figure 6d). An asymmetric supercapacitor with a V_2_O_5_/CoMoS_4_/CC/graphene positive electrode exhibited an energy density of 40.2 Wh kg^−1^ at a power density of 800 W kg^−1^.

Unlike other synthesis strategies for growing V_2_O_5_ on a carbon nanofiber substrate, the electrospinning technique can directly produce VCNF composite materials [110]. The placement of a solution of a carbonaceous material such as polyacrylonitrile (PAN) and V_2_O_5_ in an electrospinning apparatus resulted in a flexible nanofiber composite material with controllable porosity and composition [111]. Kim et al. electrospun VCNF fibers using a dispersion of PAN and different amounts of V_2_O_5_ in a DMF solution [112]. The resulting nanofibers were heat treated at 800 °C under a nitrogen atmosphere for further oxidation and stabilization. The resulting nanofiber structure was primarily composed of carbon with small, amorphous V_2_O_5_ agglomerations anchored on the CNF surface. The size of the agglomerations increased from 20 to 80 nm with an increase in the V_2_O_5_ content. Increasing the V_2_O_5_ content also increased the surface area of the electrode by introducing more pores into the nanofiber structure. The BET surface area of VCNF-20 (20 wt% V_2_O_5_) was 595.21 m^2^ g^−1^ compared to 510.45 m^2^ g^−1^ for VCNF-5 (5 wt% V_2_O_5_). In a symmetric electrode setup, the VCNF-20 electrode exhibited a C_sp_ of 150.0 F g^−1^ at a current density of 1 mA cm^−2^, which was three times greater than that of the bare CNF because of the faradic contributions from the V_2_O_5_ crystals.

Various synthesis methods are available for the fabrication of VCNF hybrid materials. Strategies that involve the coating of nanofibers on a CC substrate, such as electrodeposition, sol dipping, and crystallization, utilize the porous architecture and high conductivity of CNFs for improved electrochemical performance. Alternatively, V_2_O_5_ nanocrystals can be intercalated into the CNFs for similar benefits. Electrospinning can be used to synthesize a VCNF material without any dependence on the CNF substrate. Instead, V_2_O_5_ and the carbon precursor can be electrospun into a fibrous product, allowing easily controllable composition and porosity.

### 4.2. Effects of Composition

The ratio between CNF and V_2_O_5_ plays a vital role in determining the electrochemical properties of a VCNF composite by altering its conductivity, porosity, and morphology. Kim et al. electrospun a solution with varying V_2_O_5_ loading and PA [113]. The mixture was subsequently heat treated and steamed under a nitrogen atmosphere at 800 °C for 1 h for carbon activation. The activated VCNF material had a fibrous structure with V_2_O_5_ agglomerations of 15 to 60 nm in diameter. The size of the agglomerations increased with increasing V_2_O_5_ loading owing to a change in the viscosity and conductivity of the electrospun solution. The activation of the nanofibers increased the diameter from 108 to 200 nm. A lower V_2_O_5_ content increased the surface area of the composite with a maximum BET surface area of 1113.5 m^2^ g^−1^ for activated VCNF-5 (5 wt% V_2_O_5_) because more carbon was accessible for activation by the steam. Activated VCNF-5 had the greatest C_sp_ (73.85 F g^−1^) at a current density of 1 mA cm^−2^, which decreased slightly to 58.02 F g^−1^ at a large current density of 20 mA cm^−1^ because of its low charge transfer resistance of 1.20 Ω. The symmetric supercapacitor with activated VCNF-5 electrodes had an energy density of 68.84 Wh kg^−1^ at a power density of 20,000 W kg^−1^.

The composition of V_2_O_5_ and CNF can be altered by changing the duration of V_2_O_5_ exposure to the CNF substrate. Choudhury et al. submerged CNF paper in a solution of V_2_O_5_ and H_2_O_2_ for five and seven days [114]. The resulting free-standing paper electrode was dried at 100 °C for one day in an oven. Increased exposure to V_2_O_5_ resulted in more V_2_O_5_ crystallization on the CNFs. VCNF-5 (five days) had a V_2_O_5_ layer thickness of ~8 nm_,_ whereas VCNF-7 (seven days) had a thickness of ~17 nm. Although both VCNF variants were limited by the incomplete oxidation of V_2_O_5_, VCNF-5 exhibited higher crystallinity and decreased water intercalation because of the shorter exposure to the V_2_O_5_ solution. VCNF-5 also had a larger BET surface area of 573.65 m^2^ g^−1^ compared to the surface area of 442.16 m^2^ g^−1^ for VCNF-7 because each nanofiber in VCNF-5 was thinner, resulting in larger pore sizes. VCNF-5 exhibited larger EDL characteristics than VCNF-7 because VCNF-5 had a higher amount of accessible carbon content. VCNF-5 yielded a C_sp_ of 227 F g^−1^ at a current density of 1 A g^−1^. The C_sp_ decreased slightly to 154 F g^−1^ at a current density of 10 A g^−1^ because of its low 5.49 Ω charge transfer resistance. The VCNF-5 electrode had a maximum energy density of 63.6 Wh kg^−1^ at a power density of ~400 W kg^−1^, which decreased to 18.8 Wh kg^−1^ at a power density of 4555 W kg^−1^. Velayutham et al. examined the importance of electrodeposition duration on the composition of binderless VCNF electrodes [115]. Rope-like V_2_O_5_ with a pure orthorhombic crystal phase was evenly deposited onto the carbon fibers, resulting in V_2_O_5_ surface extrusions. The wrinkled surface from the V_2_O_5_ crystal growths resulted in a greater surface area for faradic redox reactions. Increasing the deposition time beyond 40 min resulted in a smoother surface because of the increased V_2_O_5_ growth, leading to a smaller surface area. VCNF-40 (40 min deposition) exhibited the highest areal capacitance of 394 mF cm^−2^ at a current density of 1 mA cm^−2^. A 40 min deposition time had a large V_2_O_5_ composition for higher energy storage potential while creating a rough CNF coating for more surface redox reactions. At higher scan rates, VCNF-30 performed better than VCNF-40 with an areal capacitance of 143 mF cm^−2^ at a current density of 15 mA cm^−2^ because the higher carbon composition improved the electron mobility. An asymmetric supercapacitor with a VCNF-30 positive electrode yielded an energy density of 17.7 Wh kg^−1^ at a power density of 2728 W kg^−1^.

The composition of VCNF composites can be altered by controlling the initial loading of V_2_O_5_ during synthesis or exposing a CNF substrate to V_2_O_5_ for different durations of time. Optimizing the composition of the VCNF material is essential for maximizing the faradic energy storage potential while maintaining high porosity and conductivity for the hybrid material.

### 4.3. Effects of Physical Treatment

The heat treatment of the VCNF composites improves their stability and increases V_2_O_5_ oxidation for better electrochemical performance. Chen et al. formed arrays of thin V_2_O_5_ nanosheets on a CC template via a hydrothermal synthesis route and subsequently annealed the composite at different temperatures in an H_2_/argon atmosphere [116]. An increase in the annealing temperatures created more defects in the V_2_O_5_ crystal structure, and the non-uniform crystal structure resulted in high ion diffusion and fast redox reactions. The VCNF material contained amorphous V_2_O_5_ nanosheets with more metallic than semiconductor characteristics due to the oxygen defects. Figure 7 summarizes the effects of annealing on the formation of defects for enhanced electrochemical performance. The VCNF annealed at the highest temperature of 500 °C yielded the highest areal capacitance of 554 mF cm^−2^ at a current density of 0.63 A g^−1^. An asymmetric supercapacitor with the VCNF electrode exhibited an energy density of 161.8 μWh cm^−2^ at a power density of 500 μW cm^−2^. You et al. found that annealing a VCNF composite at different temperatures affected the formation of V_2_O_5_ agglomeration [117]. A VCNF composite was synthesized via a hydrothermal process and subsequently annealed at 500 °C in a nitrogen atmosphere for different durations. V_2_O_5_ had a partial orthorhombic crystal structure owing to the incomplete oxidation of V_2_O_5_ during the hydrothermal process. Long annealing durations increased the size of the V_2_O_5_ agglomeration on the CNF surface. VCNF-24 (24 h annealing) contained a uniform distribution of V_2_O_5_ nanospheres coating the CNFs. However, an increase in the annealing time to 48 h resulted in large agglomerations of V_2_O_5_, decreasing the number of V_2_O_5_ active sites for surface redox reactions. With a homogeneous distribution of V_2_O_5_ crystals on the CNFs, VCNT-24 outperformed the other electrodes. VCNT-24 yielded a high C_sp_ of 475.5 F g^−1^ at a current density of 1 A g^−1^, more than three times that of pristine V_2_O_5_. With many V_2_O_5_ nanospheres, most of the capacitance was attributed to ion diffusion instead of surface capacitance. A flexible asymmetric supercapacitor with a VCNT-24 positive electrode exhibited a volumetric energy density of 0.928 mWh cm^−3^ at a power density of 17.5 mW cm^−3^.

Annealing the composite material also allowed the formation of doped VCNF composites. Guo et al. doped a VCNF composite with additional carbon to form carbon/V_2_O_5_ core/shell nanowires on a CNF substrate [118]. First, the VCNF electrode was fabricated via a hydrothermal technique. The VCNF was then dipped in a glucose solution and heat-treated at 500 °C under a nitrogen atmosphere, resulting in C-doped VCNF. The overall nanofibrous macrostructure of CC was preserved after heat treatment. The long CNFs were uniformly coated with thin strands of long V_2_O_5_ nanowires. The annealed glucose coated the thin V_2_O_5_ strands with an outer layer of carbon. The C-doped VCNF electrode exhibited an areal capacitance of 128.5 mF cm^−2^ at a scan rate of 10 mV s^−1^, which decreased by 60% at a scan rate of 400 mV s^−1^. The good rate capability was because of the additional carbon coating that increased the conductivity. The areal capacitance was 164.2 F cm^−2^ at a current density of 0.5 A cm^−2^, with major contributions from both faradic and EDL characteristics. The charge transfer resistance (3.8 Ω) was low because of the conductive contact between the CNFs on the CC and the outer carbon layer on the V_2_O_5_ nanowires. The carbon coating also reduced vanadium dissolution, resulting in high cyclic stability, as indicated by a high 94.4% C_sp_ retention after 10,000 cycles at a high scan rate of 100 mV s^−1^, which was significantly higher than that of pure V_2_O_5_ (13.3%). Sun et al. doped CNFs with nitrogen by first preparing a solution of CNF, pyrrole monomer, and ammonium persulfate and subsequently heating the mixture to 900 °C under argon atmosphere for 2 h, yielding an N-doped CNF (N-CNF) composite [119]. A V_2_O_5_ sol was added dropwise into the N-CNF material to yield a sol-gel that was subsequently cured for two days in an oven at 50 °C. The resulting aerogel was freeze dried and annealed at 350 °C in air, yielding a free-standing N-VCNF electrode. The N-CNFs were coated with a 20-nm-thick layer of V_2_O_5_. The V_2_O_5_ layer constituted a pure orthorhombic crystal phase, suggesting the complete oxidation of V^4+^ to V^5+^ due to the annealing procedure. The addition of the V_2_O_5_ layer to the N-CNFs decreased the BET surface area to 334.2 m^2^ g^−1^. The C_sp_ of N-VCNF was 595.1 F g^−1^ at a current density of 0.5 A g^−1^, which was almost twice that of the non-doped VCNF. The nitrogen substitutions in the CNF lattice functioned as nucleation sites for additional V_2_O_5_ crystal growth, allowing a more uniform formation of the core/shell microstructure. The homogeneous V_2_O_5_ coating decreased the charge transfer resistance to 1.04 Ω. The N-VCNF electrode exhibited good reversibility with almost no change in C_sp_ after 10,000 cycles at a current density of 0.5 A g^−1^ and only a 3% decrease after 12,000 cycles because of the stable core/shell nanostructure. The symmetrical supercapacitor with N-VCNF electrodes had a maximum energy density of 82.65 Wh kg^−1^ at a power density of 250 W kg^−1^ which decreased to 26.83 Wh kg^−1^ at a power density of 5000 W kg^−1^.

In addition to the heat treatment, other physical treatments can alter the morphology of the VCNF electrodes. Parmar et al. further modified the V_2_O_5_ crystals grown on carbon fiber surfaces using lasers to dehydrate amorphous V_2_O_5_ crystals [120]. A VCNF material with intercalated water molecules was synthesized via electrodeposition. Pristine V_2_O_5_ had an interstitial spacing of ~4.4 Å with an alternating VO_5_–VO_5_ pyramid structure, whereas hydrated V_2_O_5_ had a spacing that was two to three times greater than that of the atomic pyramids facing the same direction. De-intercalating water from the crystal structure of V_2_O_5_ decreased the interlayer gap to ~4 Å, reverting the atomic structure to the alternating pyramid form. Panigrahi et al. grew hair-like V_2_O_5_ crystals on AC felt but restrained one side of the AC felt to a glass slide [121]. Thus, V_2_O_5_ formation occurred only on the exposed side, allowing better electrical contact on the uncoated side. The acid-activated carbon felt acted as a porous template, resulting in a large BET surface area of 76.434 m^2^ g^−1^ for better ion diffusion. The porous architecture of the hair-like V_2_O_5_ resulted in a greater pseudocapacitive contributions than EDL contributions, resulting in a high C_sp_ of 460.8 F g^−1^ at a current density of 2 A g^−1^. The highly interconnected CNF structure increased the rate capability such that C_sp_ only decreased to 81.25% at 10 times the current density. The good electron mobility of the hybrid material was further supported by a low series resistance of 2.7 Ω and charge transfer resistance of 1.1 Ω. A symmetric supercapacitor with the VCNF electrode exhibited an energy density of 48.32 Wh kg^−1^ at a power density of 490 W kg^−1^.

## 5. Other V_2_O_5_/Carbon Composites

Many studies have been conducted on other V_2_O_5_/carbon (VC) composites based on carbon moieties such as carbon quantum dots (CQDs) or bio-based carbon materials. As a stable, green, and conductive material, carbon can be added as supplementary materials to enhance the pseudocapacitive performance of V_2_O_5_ for supercapacitor applications [122]. These VC hybrid materials can be synthesized with carbon obtained from both inorganic and organic sources for diverse composite nanostructures. Herein, the effects of the synthesis process, V_2_O_5_/carbon composition, and physical treatment on the electrochemical performances of other carbon-based V_2_O_5_ composites are discussed. The morphology and electrochemical performances of other V_2_O_5_/carbon composite electrodes for supercapacitor applications reported in the literature are summarized in Table 4.

### 5.1. Effects of Synthesis Method

The final nanostructure of the VC composite material depends significantly on the initial carbon precursor. Daubert et al. compared the morphologies of the VC electrodes synthesized using either microporous Supra 50 or mesoporous G60 carbon powders [123]. Both carbon powders were activated with concentrated acid and pasted on Ni foil with an acetylene black filler and PVDF binder. V_2_O_5_ was subsequently coated on the carbon-coated Ni foam using atomic layer deposition (ALD). The resulting hybrid material had a layer of amorphous V_2_O_5_ crystals covering the carbon surface. Additional V_2_O_5_ deposition cycles resulted in increasingly uneven V_2_O_5_ coatings. For mesoporous G60, the maximum C_sp_ was 540 F g^−1^ with 25 ALD cycles and 120 F g^−1^ for 75 ALD cycles. The high capacitance was due to the even V_2_O_5_ surface coating, thereby providing a large surface area for more ion adsorption. Because the micropores of Supra 50 matched the size of the atomic layer deposition precursor, the deposition process could not effectively coat Supra 50. Moreover, the coating process blocked the formation of micropores and decreased the number of active sites for faradic redox reactions. Thus, carefully selecting the carbon precursor is essential in maximizing the VC hybrid material’s electrochemical performance.

Bio-based carbon sources have also been utilized for the synthesis of VC materials. Glucose is a promising carbon precursor for VC materials because it can act as an oxidizing agent at high temperatures and function as a conductive carbon additive [124]. Narayanan employed a facile hydrothermal synthesis method using V_2_O_5_ powder and glucose as precursors to yield VC [125]. The VC material was composed of V_2_O_5_ nanorods with glucose-derived carbon quantum dots (CQDs) uniformly scattered on the nanorod surface. Incomplete oxidation of V^4+^ to V^5+^ was observed due to partial reduction by glucose; however, the V_2_O_5_ rods still exhibited an orthorhombic crystal structure. The electrochemical performance of the VC material was tested using a three-electrode configuration with a CQD counter electrode and an Ag/AgCl reference electrode in 3 M KCl electrolyte. The C_sp_ of VC was 300 F g^−1^ at a current density of 0.5 A g^−1^, which decreased slightly to 250 F g^−1^ at a current density of 2 A g^−1^. The intercalated CQDs improved the conductivity of the device, as indicated by a low charge transfer resistance of 14.5 Ω. Moreover, the layered crystal structure of the V_2_O_5_ nanorods allowed better electron mobility via the attached CQDs. An asymmetric cell with a VC working electrode yielded a C_sp_ of 119 F g^−1^ at a current density of 1 A g^−1^. The energy density was 60 Wh kg^−1^ at a current density of 1 A g^−1^. The maximum power density was 4200 W kg^−1^ at a current density of 5 A g^−1^. Balasubramanian et al. used dextran, a naturally occurring polysaccharide, as the carbon precursor [126]. Dextran was slowly added to a solution of V_2_O_5_ and H_2_O_2_ and stirred for 4 h to allow precipitation. The precipitate was subsequently annealed at 400 °C for 2 h to yield a VC hybrid material. The composite exhibited a flower-like macrostructure with urchin-like protrusions. The soft dextran decomposed and covered V_2_O_5_ in a sharp urchin-like structure after annealing. The amorphous carbon layer increased the conductivity of VC and reduced V_2_O_5_ dissolution during charge/discharge. The flower-like architecture increased the surface area of the composite, resulting in more active sites for ion intercalation/de-intercalation, increasing the C_sp_ value to a maximum of 417 F g^−1^ at a current density of 0.5 A g^−1^. The energy density of the VC composite was 47 Wh kg^−1^ at a power density of 224 W kg^−1^.

The carbon sources used for VC synthesis can also be derived from once-living organisms. Ngom et al. fabricated VC electrodes from different strains of hibiscus flower: light red (LR), dark red (DR), and white (W) [127]. The flowers were dried in sunlight and crushed into powder. The powder was dissolved in DI water and filtered to remove the large organic residue. The V_2_O_5_ powder was added to the solution with H_2_O_2_ and subsequently heated in an autoclave for one day at 180 °C. The hibiscus-derived graphitic flakes were used as the template for V_2_O_5_ growth, resulting in the nucleation of flower-like V_2_O_5_ nanosheets. H_2_O_2_ promoted the exfoliation of the carbon flakes, resulting in a more porous structure that allowed better ion diffusion. DR-VC exhibited the largest orthorhombic crystal size, with the largest specific surface area of 3.3 m^2^ g^−1^. Due to the increased surface area and organized crystal structure, DR-VC exhibited the highest specific capacity of 99.1 mAh g^−1^. An asymmetric supercapacitor with DR-VC and AC electrodes yielded an energy density of 33.4 Wh kg^−1^ at a power density of 670 W kg^−1^. Mei et al. pyrolyzed bacterial cellulose to yield carbonized bacterial cellulose (CBC), which was reacted with V_2_O_5_ powder via a hydrothermal method [128]. The resulting composite comprised web-like carbon strands coated with thick V_2_O_5_ nanobelts with a width of ~70 nm and length of ~600 nm. The amorphous carbon strands formed a highly conductive web that prevented V_2_O_5_ aggregation. The orthorhombic crystal phase V_2_O_5_ supported the faradic redox reactions, and both EDL and pseudocapacitive characteristics contributed to charge storage. The C_sp_ was 198 F g^−1^ at a scan rate of 10 mV s^−1^, which decreased to ~75 F g^−1^ at a scan rate of 200 mV s^−1^. The CBC/V_2_O_5_ composite had a maximum C_sp_ of 281 F g^−1^ at a current density of 0.25 A g^−1^, which decreased to 65 F g^−1^ at a current density of 5 A g^−1^. The high C_sp_ at a lower current density was due to the increased pseudocapacitive contribution from V_2_O_5_. The CBC/V_2_O_5_ electrode exhibited a high cyclic stability with 97% C_sp_ retention after 1000 cycles and 87% of its initial C_sp_ after 2000 cycles at a high scan rate of 100 mV s^−1^ because of the highly conductive carbon network. The V_2_O_5_/CBC electrode was highly reversible because the carbon webs stabilized the V_2_O_5_ nanobelts.

The initial carbon precursor is vital for determining the final morphology of the VC composite. A carbon source with large pores provides a large surface area for V_2_O_5_ nucleation and more active sites for faradic redox reactions. Bio-derived carbon materials from various sources such as glucose, plants, and bacteria have organized carbon structures that can be carbonized and combined with V_2_O_5_ to improve conductivity and structural stability.

### 5.2. Effects of Composition

The ratio of the carbon component to V_2_O_5_ affects the synergetic balance between conductivity and charge capacity and can also alter the morphology of the resulting composite. Fleischmann et al. varied the ratio of V_2_O_5_ to carbon onion precursors for hydrothermal synthesis [129]. Figure 8 shows the SEM micrographs of V_2_C_7_ (2:7 V_2_O_5_ to carbon onion), V_3_C_6_ (3:6 V_2_O_5_ to carbon onion), and V_4_C_5_ (4:5 V_2_O_5_ to carbon onion). Fleischmann et al. also physically combined the carbon onions and V_2_O_5_ crystals after synthesizing each of these individually (C_5_V_4_-COMP).

Quasi-spherical V_2_O_5_ nanoflowers were grown at the nucleation sites on the carbon onions. Thus, an increase in the initial carbon content also increased the V_2_O_5_ growth. An increase in the carbon onion content resulted in a larger surface area for additional surface reactions. V_2_C_7_ (Figure 8A) was more spread out with smaller agglomerations than V_3_C_6_ (Figure 8B). Increasing V_2_O_5_ content to V_4_C_5_ (Figure 8C) produced larger V_2_O_5_ agglomerations, as indicated by the dark coloring. Physical mixing of carbon onions and V_2_O_5_ resulted in large agglomerations of vanadium oxide and carbon onions (Figure 8D), which prevented the synergy between the two materials. The various VC composites were analyzed using a two-electrode configuration with a PTFE-bound AC counter electrode and a VC working electrode with LiClO_4_ in acetonitrile electrolyte. V_3_C_6_ outperformed the other three composites because of well-integrated conductive carbon with a large surface area and sizable V_2_O_5_ growth. Zhang et al. reported an increase in conductivity with an increase in the carbon content [130]. V_2_O_5_ was combined with mesoporous carbon hollow spheres (MCHSs) in different ratios to form a suspension that was rigorously stirred and subsequently freeze dried to yield a macroporous VC composite. The hybrid material comprised a web-like V_2_O_5_ maze with sporadic MCHS nanospheres anchored to the webs. Increasing the weight percentage of MCHSs from 33% (VC-33) to 67% (VC-67) increased the number of spherical carbon agglomerations. VC-33 had sparse carbon nanospheres, whereas VC-67 formed grape-like bundles on the V_2_O_5_ webs. VC-50 (50 wt% MCHS) contained a uniform spread of graphitic carbon nanospheres on the V_2_O_5_ web. A low carbon content reduced agglomerations that allowed ion adsorption into V_2_O_5_. However, an increase in the carbon content increased the conductivity of the material. Thus, VC-50 with a uniform distribution of carbon nanospheres exhibited the highest C_sp_ of 313 F g^−1^ at a current density of 0.25 A g^−1^_._ The macroporous web structure decreased the stress due to repeated ion insertion/de-insertion, resulting in an increase in cyclability from 43% for bare V_2_O_5_ to 81% for VC-50 after 4000 cycles at a current density of 5 A g^−1^.

Control of the carbon to V_2_O_5_ ratio can also change the porosity of the VC material, thereby altering the ion diffusion capability and number of active sites for faradic redox reactions. Zhu et al. determined the effects of V_2_O_5_ loading on the porosity and performance of a VC electrode [131]. The VC composite was synthesized using a liquid-phase impregnation technique, yielding a hierarchically porous VC material. The VC material had a macroporous carbon frame with V_2_O_5_ intercalated through the pores. At a low 17.6 wt% of V_2_O_5_ (VC-18), the pores in the carbon were small and tightly packed, resulting in a BET surface area of 622 m^2^ g^−1^. At a high V_2_O_5_ loading of 52 wt% (VC-52), the sheets were covered with large micropores with tiny gaps between the layers, resulting in a smaller BET surface area of 487 m^2^ g^−1^. The VC material with a 38.7 wt% V_2_O_5_ (VC-39) had the greatest BET surface area of 645 m^2^ g^−1^ because of its larger pores. Thus, VC-39 yielded the highest C_sp_ of 492.1 F g^−1^ at a scan rate of 5 mV s^−1^ because of the numerous active sites provided by the micropores and ease of ion diffusion through the high volume of macropores. The C_sp_ value decreased slightly to ~400 F g^−1^ at a scan rate of 100 mV s^−1^, indicating good rate capability because the amorphous carbon dispersion enabled good electron mobility with low series and charge transfer resistances of 0.54 and 1.05 Ω, respectively. A symmetric supercapacitor assembled with VC-39 had an energy density of 87.6 Wh kg^−1^ at a power density of 497 W kg^−1^ that decreased to 20.4 Wh kg^−1^ at a power density of 3272 W kg^−1^. Saravanakumar et al. controlled the ratio of N-doped mesoporous carbon (N-MPC) nanospheres to V_2_O_5_ to obtain a superior supercapacitor electrode with maximum porosity [132]. The VC material comprised V_2_O_5_ flakes anchored onto the spherical N-MPC. These V_2_O_5_ covered carbon spheres formed agglomerations with other VC nanospheres. The hydrothermal method oxidized V_2_O_5_ to form pure orthorhombic crystals, while reducing the carbon into a graphitic crystal phase. A low N-MPC loading of 10% (VC-10) resulted in many uncoated carbon spheres due to V_2_O_5_ agglomeration. This decreased the accessibility to V_2_O_5_ for surface ion intercalation. At 5 wt% N-MPC, the carbon nanospheres were homogeneously covered to maximize the pseudocapacitive capacity of V_2_O_5_ with an optimal surface area of 8.77 m^2^ g^−1^. The addition of 15 wt% N-MPC resulted in nanospheres that were completely coated by V_2_O_5_, decreasing the accessibility to N-MPC for a suboptimal surface area of 10.35 m^2^ g^−1^. At 10 wt% N-MPC, the carbon nanospheres were effectively covered to maximize the pseudocapacitive capacity of V_2_O_5_ with an optimal surface area of 8.77 m^2^ g^−1^. VC-10 exhibited the highest C_sp_ of 487 F g^−1^ at a current density of 0.5 A g^−1^, which was 34% higher than that of bare V_2_O_5_. The increase in capacitance was caused by a significant increase in surface area from 5.64 to 8.77 m^2^ g^−1^ and increased conductivity because of N-MPC. An asymmetric supercapacitor with the VC-10 electrode had an energy density of 12.8 Wh kg^1^ at a power density of 317 W kg^−1^. Kudo et al. added varying amounts of acetylene black to a V_2_O_5_ sol to create a suspension with acetone surfactant [133]. The Ni foam was subsequently submerged in the suspension and heated at 120 °C for 5 h. The resulting VC was comprised of spherical carbon cores with an uneven V_2_O_5_ outer layer coating. These nanospheres formed bumpy agglomerations, yielding a BET surface area of 30 m^2^ g^−1^. With a large carbon content (30 wt% acetylene black), the VC exhibited good rate capability, maintaining a charge capacity of 340 mAh g^−1^ at a high current density of 54 A g^−1^. The VC electrode maintained 100% of its capacity even after 2000 cycles at a charge rate of 20 C because the porosity of the VC material allowed unobstructed Li ion insertion/de-insertion. The maximum energy density was 80 Wh kg^−1^ at a power density of 26,000 W kg^−1^ with an average working voltage of 2 V, which rapidly deteriorated with repeated cycles. The average energy density was 15–20 Wh kg^−1^. Peng et al. followed a similar synthesis route using Ketjen black powder and a V_2_O_5_ sol [134]. Ni foam was used as a template for VC coating. The VC material comprised V_2_O_5_ nanosheets covered with carbon nanospheres uniformly distributed on the hydrated V_2_O_5_ surface. An increase in the initial Ketjen black loading increased the final surface area of VC, with a maximum BET surface area of 264.72 m^2^ g^−1^ for VC_1_ (1 g Ketjen black loading). VC_0.5_ (0.5 g Ketjen black loading) exhibited the highest C_sp_ of 1634 F g^−1^ at a current density of 5 mA cm^−2^ because of effective synergy between carbon for fast electron mobility and V_2_O_5_ nanosheets for high pseudocapacitance. A symmetric supercapacitor with the VC_0.5_ electrodes had an energy density of 56.83 Wh kg^−1^ at a power density of 303 W kg^−1^, which decreased slightly to 30.86 Wh kg^−1^ at a power density of 2433 W kg^−1^. Despite not having a lower surface area and containing less conductive carbon content than VC_1_, VC_0.5_ retained a good rate capability while sustaining a high energy storage capacity.

### 5.3. Effects of Physical Treatment

Heat treatment of a VC composite can alter the morphology of the hybrid material [135]. Kim et al. synthesized VC by heating a mixture of vanadium trichloride, terephthalic acid, and DI water in an autoclave at 200 °C for four days. Subsequently, the VC powder was calcinated at 400 °C for 6 h in an argon atmosphere [136]. Prior to calcination, the VC exhibited a rectangular structure with large crystals that were 1–4 μm in length. The heat-treated VC had a highly orthorhombic crystal structure with thinner nanorods that were 0.5–3 μm in length. A 4–6-nm-thick layer of graphitic carbon coated the V_2_O_5_ nanorods. The annealed VC exhibited an initial discharge rate of 286 mAh g^−1^ at 0.1 C, which was close to the theoretical limit of 294 mAh g^−1^. Zhang et al. also analyzed the effects of annealing temperature and duration on the morphology of the VC material [137]. The VC composite was synthesized via a hydrothermal method and was further heat treated in a muffle furnace at varying temperatures of 300 to 500 °C for different durations in air. The hydrothermal synthesis yielded smooth prism-like monoclinic V_2_O_5_ protrusions with amorphous carbon nanospheres embedded between the V_2_O_5_ crystals. Calcination under air resulted in porous V_2_O_5_ nanoparticles with uneven surfaces due to combustion reactions with air. At a low calcination temperature of 300 °C, many carbon nanospheres remained scattered within the V_2_O_5_ nanorods, whereas no nanospheres were present in the VC that was heat treated at 500 °C. The presence of carbon nanospheres indicated the incomplete oxidation of V_2_O_5_, resulting in a lower C_sp_ of 151 F g^−1^ at a current density of 1 A g^−1^ for VC-300 compared to 367 F g^−1^ for VC-500. An increase in the calcination duration up to 8 h increased the C_sp_ of the VC composite because V_2_O_5_ had more time to nucleate. However, further increasing the calcination time to 12 h decreased the C_sp_ because the agglomeration of V_2_O_5_ resulted in limited surface ion diffusion.

Heat treatment can also promote the oxidation of V^4+^ to V^5+^ for a more crystalline V_2_O_5_. Zeiger et al. fabricated vanadium carbide/carbide core/shell composites (VC-C) by reacting vanadium carbide with NiCl_2_ in a graphite crucible at 700 °C in chlorine gas followed by calcination at 450 °C in synthetic air to promote further oxidation of V^4+^ [138]. The resulting VC-C nanostructure was composed of a V_2_O_5_ core and carbide-derived carbon shell. Increasing the calcination temperature to 600 °C increased V_2_O_5_ oxidation but burned off most of the carbon shell. VC-C-90 (90% theoretically converted) exhibited the highest specific capacity of 415 mAh g^−1^ at a current density of 0.01 A g^−1^ with almost 100% coulombic efficiency. The partial conversion of the V_2_O_5_ core from vanadium carbide provided a large storage capacity, while the outer carbon shell increased the conductivity of the composite. A decrease in the annealing temperature to decrease the carbide shell burnoff while effectively oxidizing the vanadium carbide core was essential for maximum yield. An asymmetric supercapacitor setup with the VC-C-90 negative electrode yielded an energy density of 90 Wh kg^−1^ for charging and 50 Wh kg^−1^ for discharging at a power density of 166 W kg^−1^. The energy density decreased to 27 Wh kg^−1^ at a high-power density of 6700 W kg^−1^, retaining 80% of its energy density after 10,000 cycles at a current density of 1 A g^−1^. Narayanan et al. annealed glucose-based VC at temperatures of 250–400 °C in air [139]. The composite contained thin V_2_O_5_ nanorods with CQDs anchored to the nanorods; the V_2_O_5_ nanorods also had a thin carbon coating. Increasing the annealing temperature decreased the thickness of the V_2_O_5_ nanorods and promoted V_2_O_5_ oxidation using oxygen from the atmosphere, resulting in more orthorhombic V_2_O_5_ crystals. VC annealed at 250 °C (VC250) exhibited the highest C_sp_ of 260 F g^−1^ at a current density of 1 A g^−1^. The layered V_2_O_5_ crystal structure improved electron propagation, and the high surface area of the VC material increased the surface adsorption. VC250 also exhibited the lowest charge transfer resistance of 11.4 Ω. The VC250 electrode exhibited high cycling capability, retaining 92% of its C_sp_ after 5000 cycles at a current density of 5 A g^−1^ because the carbon coating decreased vanadium dissolution. The imperfections in the V_2_O_5_ lattice caused by V^4+^ enhanced the Li-ion intercalation capability and decreased V_2_O_5_ dissolution.

The heat treatment of VC composites can change the morphology of the composite and promote the oxidation of vanadium oxide to V_2_O_5_. Thus, controlling the annealing temperature and duration allows the formation of partially oxidized VC materials for increased stability and surface ion intercalation. However, excessive annealing temperatures and durations disintegrate the carbon content, resulting in a low conductivity. Therefore, it is essential to optimize both the annealing temperature and duration.

## 6. Conclusions

The synthesis of various metal oxide and carbon composites as supercapacitor electrode materials has attracted significant attention because of the increasing demand for high energy and power-dense energy storage devices used in portable electronics and electric vehicles. The development of stable vanadium oxide-based electrode materials has been extensively investigated due to the high energy storage potential and natural abundance of vanadium oxide. Common carbon nanostructures such as rGO, CNTs, CNFs, and CQDs have been combined with V_2_O_5_ to yield high-performance supercapacitor electrode materials.

V_2_O_5_/rGO composites can be synthesized using many methods, including hydro/solvothermal, sol-gel, filtration, and chemical deposition methods. These strategies often yield a lamellar rGO nanostructure with V_2_O_5_ intercalated between the nanosheets. V_2_O_5_/CNT hybrid materials are frequently synthesized as vertically aligned nanotubes infiltrated by V_2_O_5_ crystals or long nanotubes with V_2_O_5_ growth using various synthesis methods. The CNTs have the advantage of facile functionalization with hydroxyl or carboxyl groups to increase V_2_O_5_ nucleation and conductivity. The V_2_O_5_/CNF composites offer the unique advantage of having a highly stable 3D macrostructure that can be used as a template for the fabrication of free-standing and binderless electrodes. Using electrodeposition, crystallization, sol-dipping, or electrospinning methods, a fibrous structure with a CNF/V_2_O_5_ core/shell nanostructure or V_2_O_5_ crystals intercalated into web-like CNFs can be synthesized. Other V_2_O_5_/carbon materials such as CQD-based and amorphous carbon-flake-based composites can be easily synthesized via facile strategies such as the hydro/solvothermal method. Carbon precursors can be sourced from both artificially synthesized and bio-based carbonaceous materials.

In most cases, an increase in the V_2_O_5_ content in the V_2_O_5_/carbon hybrid materials increases the total energy storage potential because of faradic redox reactions. However, a high V_2_O_5_ content results in agglomerations that hinder surface ion adsorption. An increase in the carbon content leads to increased EDL contributions and conductivity for improved rate capability and cyclic stability. However, the low energy storage potential of carbon nanomaterials limits high carbon content for energy-dense electrode materials. An optimal balance of V_2_O_5_ and carbon can increase the surface area and porosity beyond that of solely the carbon material alone, enabling more activation sites for greater ion intercalation/de-intercalation. The synergy between V_2_O_5_ and carbon can also inhibit vanadium ion dissolution, resulting in a more stable charge/discharge. Physical treatment via annealing, calcination, and laser treatment can promote vanadium oxide oxidation to V_2_O_5_, altering the crystallinity of the hybrid material for improved electrochemical performance.

This review discusses the effects of different synthesis methods, V_2_O_5_/carbon compositions, and physical treatment strategies on the morphology and electrochemical performances of V_2_O_5_/carbon composites. This review is expected to serve as a catalyst for further research for the development of an ideal supercapacitor electrode material with high power and energy properties. Furthermore, light, solid-state supercapacitors based on V_2_O_5_/carbon nanomaterials have potential applications for portable, stretchable and wearable electronics. Continued research efforts in this area could make great contribution to developing supercapacitor technologies.

## Figures and Tables

**Figure 1 nanomaterials-11-03213-f001:**
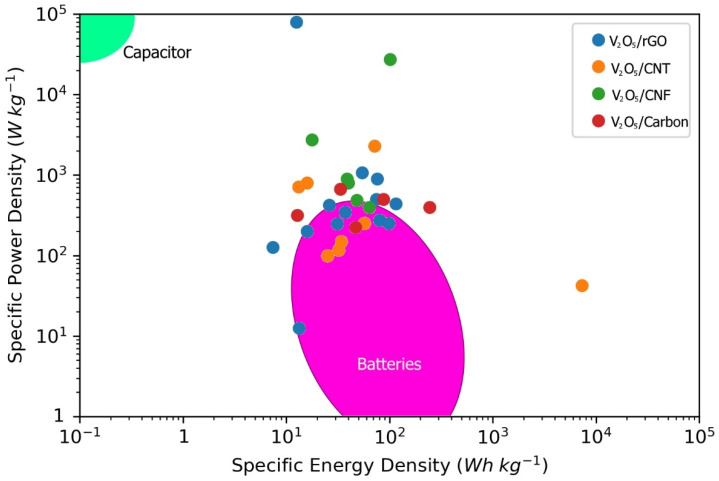
Ragone plot showing the energy and power density ranges of common electrochemical energy storage devices such as capacitors, supercapacitors, and batteries.

**Figure 2 nanomaterials-11-03213-f002:**
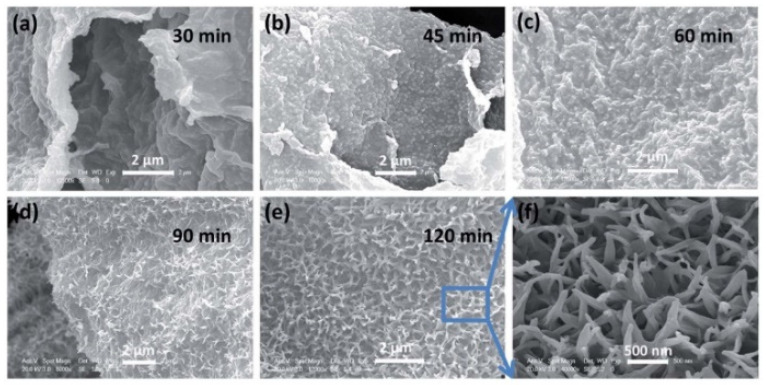
SEM images of N-doped rGO anchored to vertically aligned V_2_O_5_ nanowires after solvothermal synthesis for (**a**) 30, (**b**) 45, (**c**) 60, (**d**) 90, and (**e**) 120 min. (**f**) Magnified SEM image of the N-doped VrG after solvothermal reaction for 120 min. Reprinted with permission from Ref. [57]. Copyright 2018 Royal Society of Chemistry.

**Figure 3 nanomaterials-11-03213-f003:**
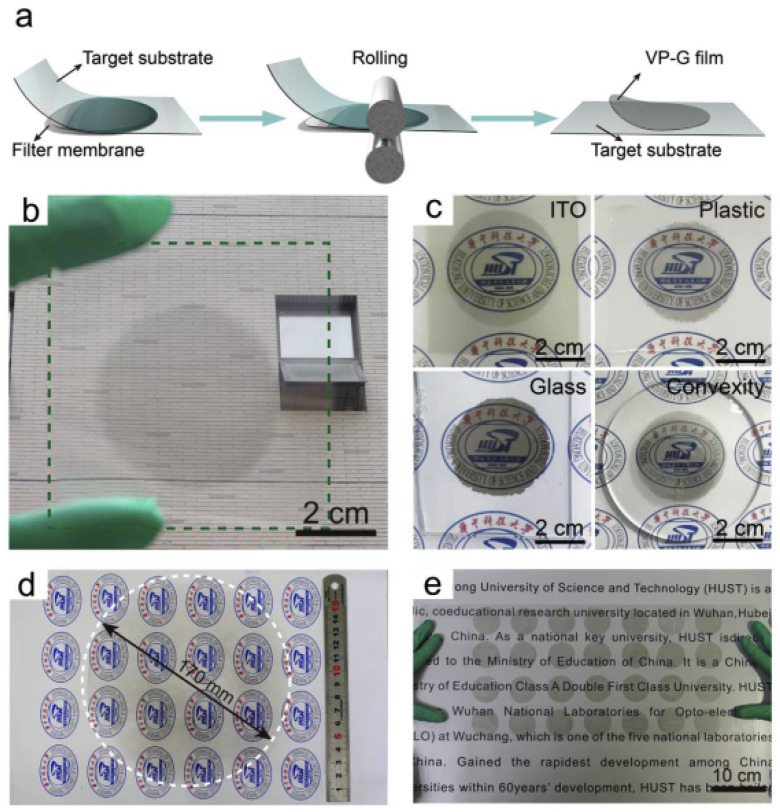
(**a**) Illustration of the transfer of the VrG film onto a substrate via a rolling process. Digital photographs of (**b**) transparent VrG films with a thickness of 22 nm, (**c**) transparent VrG on ITO, plastic, and glass with good convexity, and (**d**) a large VrG disc pressed on a substrate. (**e**) Digital photograph of a 7 × 4 array of the large-scale VrG films. Reprinted with permission from Ref. [64]. Copyright 2020 Elsevier.

**Figure 4 nanomaterials-11-03213-f004:**
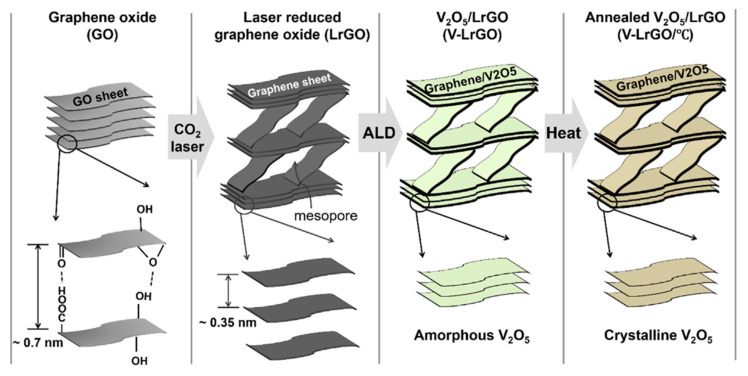
Schematic illustration for the fabrication process of the multilayered graphene (i.e., LrGO) and amorphous or crystalline V_2_O_5_ coated LrGO. Firstly, the prepared GO was reduced by a home engraver equipped with a CO_2_ laser. Then, the LrGO was coated with amorphous V_2_O_5_ by a low-temperature ALD process, followed by annealing at high temperatures. Reprinted with permission from Ref. [82]. Copyright 2020 Elsevier.

**Figure 5 nanomaterials-11-03213-f005:**
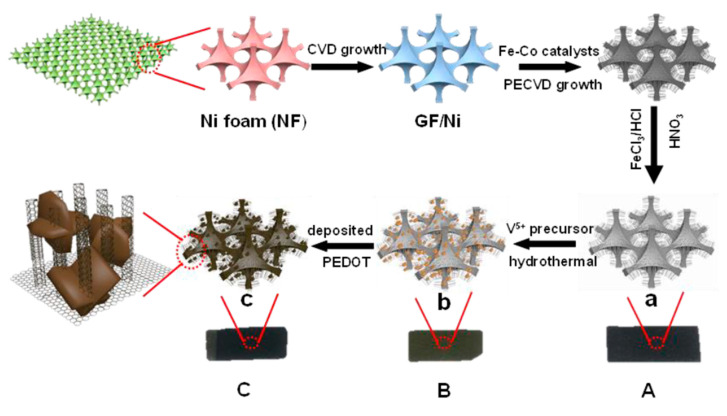
Schematic of the synthesis of (**A**) CNT/GF, (**B**) VCNT/GF, (**C**) VCNT/GF/PEDOT. Reprinted with permission from Ref. [88]. Copyright 2017 Royal Society of Chemistry.

**Figure 6 nanomaterials-11-03213-f006:**
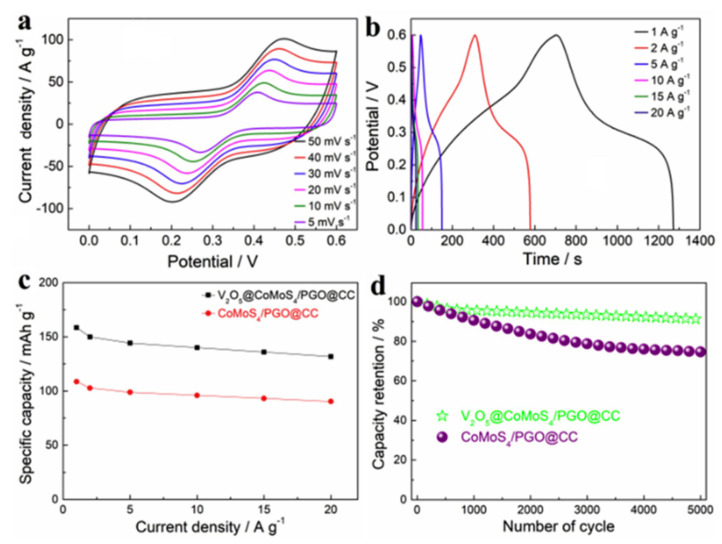
(**a**) CV and (**b**) GCD curves of V_2_O_5_/CoMoS_4_/CC/graphene at different scan rates and current densities, respectively. (**c**) C_sp_ values of V_2_O_5_/CoMoS_4_/CC/graphene and CoMoS_4_/CC/graphene at different current densities and (**d**) cycling performances of V_2_O_5_/CoMoS_4_/CC/graphene and CoMoS_4_/CC/graphene. Reprinted with permission from Ref. [109]. Copyright 2020 Elsevier.

**Figure 7 nanomaterials-11-03213-f007:**
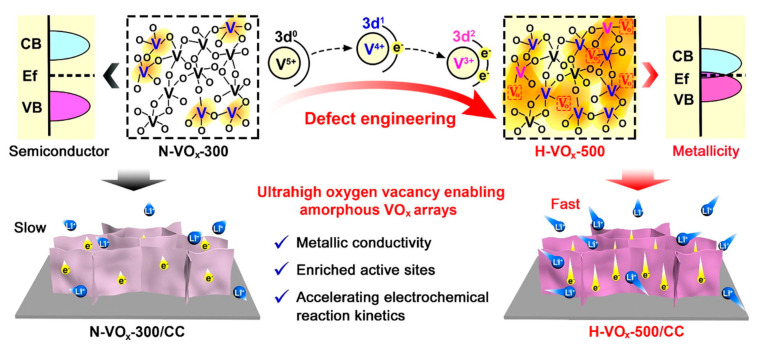
Schematic of the regulation of the V 3d band edge with high electrical conductivity by defect engineering and electrochemical advantages of the as-formed metallic amorphous VO_x_ nanosheet arrays. Reprinted with permission from Ref. [116]. Copyright 2021 Elsevier.

**Figure 8 nanomaterials-11-03213-f008:**
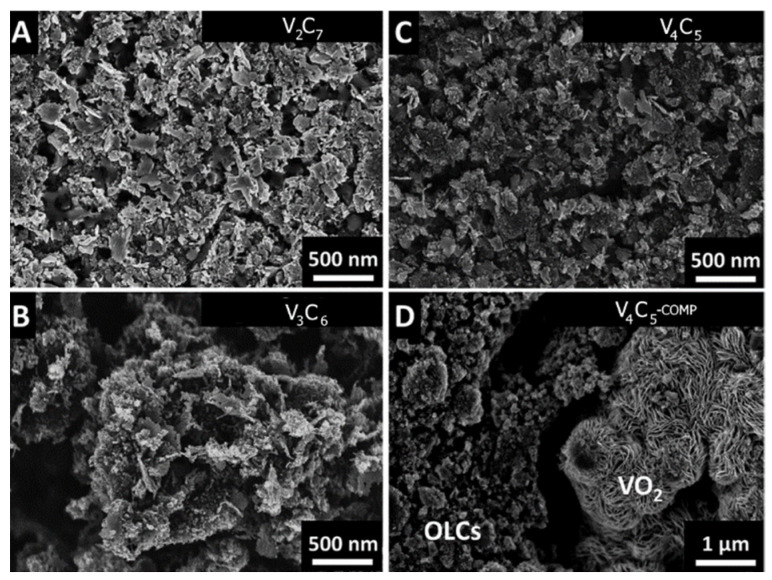
SEM micrographs of (**A**) V_2_C_7_, (**B**) V_3_C_6_, (**C**) V_4_C_5_, and (**D**) V_4_C_5_-COMP. Reprinted with permission from Ref. [129]. Copyright 2017 Royal Society of Chemistry.

**Table 1 nanomaterials-11-03213-t001:** V_2_O_5_ morphology and electrochemical performances of V_2_O_5_/reduced graphene oxide composite electrodes for supercapacitor applications.

Morphology	Maximum C_sp_ (F g^−1^)	Cycling C_sp_ Retention (%)	Cycle Number	Energy Density (Wh kg^−1^)	Power Density (W kg^−1^)
Nanowires	579	79	5000	-	-
Nanowires	710	95	20,000	98.6	250
Nanosheets	635	94 *	3000 *	75.9	900
Nanostrips	309	95.2	10,000	475	-
Amorphous	178.5	85	8000	13.3	12.5
Nanoflowers	1235	92	5000	116	440
Nanoribbons	-	720	500	16	200
Nanorods	37.2	90 *	1000 *	54.2	1075.9
Amorphous	484	83	1000	7.4	127
Amorphous	226	92	5000	12.5	79,900
Nanobelts	128.8	82	5000	-	-
Nanobelts	310.1	90.2 *	5000 *	31.3	249.7
Nanofibers	218	87 *	700 *	37.2	345
Nanowires	272	80	1000	26.22	425
Nanorods	537	84	1000	74.58	500
Nanospheres	386	-	-	80.4	275

* Cycling performance was determined using a two-electrode configuration.

**Table 2 nanomaterials-11-03213-t002:** V_2_O_5_ morphology and electrochemical performances of V_2_O_5_/carbon nanotube composite electrodes for supercapacitor applications.

Morphology	Maximum C_sp_ (F g^−1^)	Cycling C_sp_ Retention (%)	Cycle Number	Energy Density (Wh kg^−1^)	Power Density (W kg^−1^)
Nanostars	1016	64 *	5000 *	13.24	710
Nanospheres	125	-	400	25	100
Nanobelts	685	99.7	10,000	34.3	150
Amorphous	410	86	600	57	250
Nanoflakes	629	93	4000	72	2300
Nanosheets	207.7	75 *	10,000 *	7300	42.6
Nanospheres	284	76 *	5000 *	32.3	118
Nanosheets	357.5	99.5	1000	-	-
CNT Coating	510	96	5000	16	800

* Cycling performance was determined using a two-electrode configuration.

**Table 3 nanomaterials-11-03213-t003:** V_2_O_5_ morphology and electrochemical performances of V_2_O_5_/carbon nanofiber composite electrodes for supercapacitor applications.

Morphology	Maximum C_sp_ (F g^−1^)	Cycling C_sp_ Retention (%)	Cycle Number	Energy Density (Wh kg^−1^)	Power Density (W kg^−1^)
Nanorods	535.3	91.1	5000	38.7	900
CNF Coating	-	-	-	101	27,370
Nanosheets	-	89.3	10,000	40.2	800
CNF Coating	227	89	2000	63.6	400
CNF Coating	-	94 *	10,000 *	17.7	2728
Nanospheres	475.5	89.7 *	6000 *	-	-
Nanohairs	460.8	92 *	10,000 *	48.32	490

* Cycling performance was determined using a two-electrode configuration.

**Table 4 nanomaterials-11-03213-t004:** V_2_O_5_ morphology and electrochemical performances of other V_2_O_5_/carbon composite electrodes for supercapacitor applications.

Morphology	Maximum C_sp_ (F g^−1^)	Cycling C_sp_ Retention (%)	Cycle Number	Energy Density (Wh kg^−1^)	Power Density (W kg^−1^)
Amorphous	120	89	10,000	-	-
Nanorods	300	87 *	5000 *	-	-
Nanoflowers	417	92.3	2250	47	224
Nanoflowers	-	88 *	25,000 *	33.4	670
Nanobelts	281	87	2000	-	-
Nanorods	313	81	4000	-	-
Amorphous	157.7	93 *	10,000 *	87.6	497
Nanosheets	487	84	2000	12.8	317
Nanobelts	406	13.8	100	245.7	396
Nanobelts	260	92	5000	-	-

* Cycling performance was determined using a two-electrode configuration.

## Data Availability

Not applicable.

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
