# Peer review of "Recent Development in Vanadium Pentoxide and Carbon Hybrid Active Materials for Energy Storage Devices"

_nanomaterials, 2021, doi:10.3390/nano11123213_

Round 1

Reviewer 1 Report

This review manuscript summarized the current progresses of Vanadium Pentoxide and various carbon nanomaterials, such as graphen oxide, carbon nanotube,
carbon nanofibers in energy storage.The originality and the accuracy of the manuscript are good. As a result, this review paper is instructive.
According to the manuscript, there are some suggestions needed to be taken into consideration.
1) There is only conclusiong in the end of this manuscript, where the progress of V2O5@C is summarized again. As a review paper, the outlook or perspective
is essentional, so that readers can obtain more informations on the future study in the related research.
2) English may need to improve, especially for articles, prepositions, conjunctions in the context.
For example, The large surface area improves the contact between the electroactive material and the current collector for more efficient electron transfer.
How does the surface area improve the contact? what's the logic relationship between the surface area and contact?
3) In the section 2,3.4,5, there are almost the same subtitles, which seems boring and repetity. Can the contents be reorganized more interesting?

Author Response

Reviewer #1

This review manuscript summarized the current progresses of Vanadium Pentoxide and various carbon nanomaterials, such as graphen oxide, carbon nanotube, carbon nanofibers in energy storage.The originality and the accuracy of the manuscript are good. As a result, this review paper is instructive. According to the manuscript, there are some suggestions needed to be taken into consideration.

1) There is only conclusiong in the end of this manuscript, where the progress of V2O5@C is summarized again. As a review paper, the outlook or perspective is essentional, so that readers can obtain more informations on the future study in the related research.

Response

On page 35, we have added the following descriptions as suggested.

“This review discusses the effects of different synthesis methods, V2O5/carbon compositions, and physical treatment strategies on the morphology and electrochemical performances of V2O5/carbon composites. This review is expected to serve as a catalyst for further research for the development of an ideal supercapacitor electrode material with high power and energy properties. Furthermore, light, solid-state supercapacitors based on V2O5/carbon nanomaterials have potential applications to portable, stretchable and wearable electronics. Continued research efforts in this area could make great contribution to developing supercapacitor technologies.”

2) English may need to improve, especially for articles, prepositions, conjunctions in the context. For example, The large surface area improves the contact between the electroactive material and the current collector for more efficient electron transfer. How does the surface area improve the contact? what's the logic relationship between the surface area and contact?
Response

We have double checked all the sentences to improve the English.

On page 3, we have revised as follows. “The large surface area improves the interfacial contact between the electroactive material and the current collector, resulting in more efficient electron transfer.”

3) In the section 2,3.4,5, there are almost the same subtitles, which seems boring and repetity. Can the contents be reorganized more interesting?

Response

Our review paper deals with four kinds of V2O5/carbon materials. The section #2, #3, #4 and $5 introduce V2O5/rGO, V2O5/CNT, V2O5/CNFs and other V2O5/carbon composites, respectively. In particular, we investigated in detail Effects of Synthesis Method, Effects of Composition and Effects of Physical Treatment using the subsection. Thus, we believe our reviewer paper is well organized and will be of interest to its general readership.

Reviewer 2 Report

This is a nice, clear paper with many experimental details and supporting conclusions.
In my opinion the list of references is complete. Maybe the ms. is slightly too bulky. I have only some minor suggestions to the authors:

  1. Tables 1 and 2 are not referred to in the text. Also, the table labels do not give sufficient information.
  2. Table 3, as above; composition of the materials should be given. Also the first column species are not clearly distanced.
  3. Table 4, as 1.

Author Response

Reviewer #2

This is a nice, clear paper with many experimental details and supporting conclusions. In my opinion the list of references is complete. Maybe the ms. is slightly too bulky. I have only some minor suggestions to the authors:

  1. Tables 1 and 2 are not referred to in the text. Also, the table labels do not give sufficient information.

Response

As suggested, we have added the following descriptions.

On page 4, “The morphology and electrochemical performances of V2O5/rGO electrodes for supercapacitor applications reported in the literature are summarized in Table 1.

On page 16, “The morphology and electrochemical performances of V2O5/CNT electrodes for supercapacitor applications reported in the literature are summarized in Table 2.”

On page 22, “The morphology and electrochemical performances of V2O5/CNF electrodes for supercapacitor applications reported in the literature are summarized in Table 3.”

On page 28, “The morphology and electrochemical performances of other V2O5/carbon composite electrodes for supercapacitor applications reported in the literature are summarized in Table 4.”

  1. Table 3, as above; composition of the materials should be given. Also the first column species are not clearly distanced.

Response

As suggested, we have revised them.

  1. Table 4, as 1.

Response

As suggested, we have revised them.